



# Comparing an idealized deterministic-stochastic model (SUP model, version 1) of the tide-and-wind driven sea surface currents in the Gulf of Trieste to HF Radar observations

Sofia Flora[1,2], Laura Ursella[2], and Achim Wirth[3]

[1]University of Trieste, Department of Mathematics, Informatics and Geosciences, Trieste, Italy
[2]National Institute of Oceanography and Applied Geophysics - OGS, Trieste, Italy
[3]Univ. Grenoble Alpes, CNRS, Grenoble INP, LEGI, 38000 Grenoble, France

**Correspondence:** Sofia Flora (sflora@ogs.it)

**Abstract.** In the Gulf of Trieste the sea surface currents are observed by High Frequency Radar for almost two years (2021-2022) at a temporal resolution of 30 min. We developed a hierarchy of idealized models to simulate the observed sea surface currents, combining a deterministic and a stochastic approach, in order to reproduce the externally forced motion and the internal variability, which is characterized by a fat-tailed statistics. The deterministic signal includes tidal and Ekman forcing
and resolves the slowly varying part of the flow, while the stochastic signal represents the fast-varying small-scale dynamics, characterized by Gaussian or fat-tailed statistics, depending on the statistical used. This is done using Langevin equations and modified Langevin equations with a Gamma distributed variance parameter. The models were adapted to resolve the dynamics under nine tidal and wind Forcing Protocols in order to best fit the observed forced motion and internal variability Probability Density Function (PDF). The stochastic signal requires 2 stochastic degrees of freedom when the averaged tidal forcing is
adopted, while it needs 1/2 stochastic degree of freedom when the complete tidal forcing is used. Despite its idealization, the deterministic-stochastic model with stochastic fat-tailed statistics captures the essential dynamics and permits to mimic the observed PDF. Moreover, a Fluctuation Response Relation is valid when the stochastic signal is perturbed, showing that the response to an external perturbation can be obtained by considering the fluctuations of the unperturbed system.

# 1 Introduction

The Gulf of Trieste is a shallow, semi-enclosed basin in the northern Adriatic Sea (Mediterranean Sea, Fig. 1). Its surface circulation is influenced by the broader Adriatic cyclonic (counter-clockwise) circulation and the basin's thermohaline stratification and circulation (Cosoli et al., 2012; Querin et al., 2021). Due to its shallowness and small scale (horizontal length scale of around 20 km and depth of around 25 m), its current dynamics highly depend on two external forcings: the wind forcing and
the Isonzo/Soča river freshwater input (Querin et al., 2006; Cosoli et al., 2012, 2013; Querin et al., 2021).





The Bora and the Sirocco are the main strong wind patterns affecting the dynamics in the Gulf of Trieste. The Bora is an East-North-Easterly katabatic wind, bringing cold and dry air from the continent over the sea (Poulain and Raicich, 2001). Its strength is heavily influenced by local topography, making it particularly powerful over the Gulf of Trieste where it can reach gusts up to 180 km/h. When it blows, surface water is pushed out of the Gulf towards the Adriatic basin, and a compensating
bottom counter-current flows in, causing upwelling on the coastal side (Malačič et al., 2001; Querin et al., 2006; Reyes Suárez et al., 2022). The Sirocco is a moist, warm and relatively mild southerly wind, channeled by the Adriatic coastal mountains. It causes sea level rise in the northern Adriatic, leading to a strong southward return flow when the wind diminishes (Cosoli et al., 2012).

In recent years, the Gulf of Trieste has benefitted from an abundance of observational data, particularly High Frequency Radar (HFR) sea surface currents. The amount of data will strongly increase in the near future. This large amount of data allowed for a statistical approach to characterize the data (Flora et al., 2023). In particular, the authors found the analytical Probability Density Function (PDF) of the observed HFR sea surface velocity increments using a superstatistical (a superposition of statistics) analysis and the principle of maximum entropy. Superstatistical analysis (Beck et al., 2005) considers a system with a clear
time scale separation: at fast time scale a Gaussian PDF is observed of which the variance evolves slowly, on the long time scale. In the Gulf of Trieste the fast time scale of around 2 h is consistent with the turbulence of the mixed layer dynamics. The slow time scale of almost 2 days is consistent with the variability of the synoptic wind forcing, and the variance is distributed according to a Gamma distribution with shape parameter equal to 2. The Gaussian is the minimum entropy PDF with a finite variance and the Gamma distribution is the minimum entropy PDF for a positive variable, as is the variance, with a specified
mean. The total PDF of the sea surface velocity increments is a fat-tailed PDF, that means that extreme events occur more often than in a Gaussian statistics. The superstatistical analysis however is descriptive and has no predictive skills.

When modelling the turbulent ocean currents subject to a known wind forcing, there is part of the dynamics which can be described by a deterministic model and the unresolved turbulent processes that have to be described stochastically. This con-
cept is reflected in Stochastic Differential Equations (SDEs) and the corresponding Fokker-Planck equations. The former consider the evolution of many possible realizations which allow to construct a time evolving PDF, while the latter describe the deterministic evolution of the PDF. The use of SDEs in air-sea interaction was pioneered by Hasselmann (1976), who initiated a strand of scientific research on stochastic climate models. Taking into account the coupled ocean-atmosphere-cryosphere-land system, he divided it into a fast varying "weather" system and a slowly responding "climate" system, clearly separated
by a fast and a slow time scale. In his modelization the slow climate variables play the role of large particles interacting with an ensemble of smaller particles, the fast weather variables in the analogy, of the so-called Brownian motion problem. In this picture, the dispersion of the climate variables is inferred from the statistics of the weather variables with which they interact. Brownian motion was first analytically described by Einstein (1905) and Langevin et al. (1908) (see Einstein (1956) and Lemons and Gythiel (1997) respectively for an English translation). Since that time, the SDE of the motion of a Brownian par-
ticle is called the "Langevin equation". The SDE approach, adopted also in this paper, has been widely used in the past decades





in a variety of scientific fields: statistical mechanics (Baldovin et al., 2018), condensed matter physics (Silveira and Aarão Reis, 2012), marine biology and oceanography (Brillinger and Stewart, 2010), space weather science (Alberti et al., 2018), biophysics (Ham et al., 2022) and finance (Wand et al., 2024) to name a few. Regarding climate science, the Langevin equation has been widely used (Berglund and Gentz, 2002; Wirth, 2019; van den Berk et al., 2021) and great improvements have been achieved. Franzke et al. (2015) and Palmer (2019) show that including stochasticity into the parameterized representations of subgrid processes of physical climate systems has improved the skill of forecasts and reduced systematic model error. It is important to remark that the use of additive noise in linear SDE has limitations in terms of extreme events predictability, since it leads to a Gaussian statistics. Adopting multiplicative noise or nonlinear damping, taking into account non-linear interactions between resolved and unresolved modes of variability (Franzke et al., 2005; Wirth, 2018) leads to non-Gaussian statistics increasing the fatness of the PDF tails of the modelled variables (Sura, 2013).

Modern concepts of non-equilibrium statistical mechanics have been applied to environmental fluid dynamics and to components of the climate system. Often these considerations are limited to conceptual pertinence and tested on idealized models, such as the Lorentz models (Lorenz, 1996) and others, or on data from numerical models. In the present work we apply a hierachy of idealized models for the sea surface current observations from the Gulf of Trieste, starting with a deterministic modelization and then adding increasingly complex stochasticity. For solving these models we rely on analytical calculations where possible and proceed with solutions of involved SDEs. This approach enables connecting observational data in a systematic way to the underlying principles of physics.

Our modelization is idealized and local: it considers one point in the space, evolving in time, with wind shear and tidal deterministic forcing and a stochastic signal that mimics all the smaller unresolved scale dynamics. The aim of the modelization is to simulate the evolution of the forced motion and of the observed analytical PDF. It is based on the slow evolution of a short time scale Gaussian and can be interpreted by a SDE. This enables us to understand which is the role of the stochasticity in the simulation of the Gulf of Trieste wind-and-tide-driven circulation. The model is finally used to test the Fluctuation Response Relation (FRR), described by Lacorata and Vulpiani (2007), that relates the system's reaction to external perturbations to spontaneous fluctuations of the unperturbed system. An evaluation of the ocean currents forecasting based on the SDE predictability is also given.

In Sect. 2 the observed HFR sea surface data and the model forcing atmosphere data are presented. In Sect. 3 the modelization is explained with some FRR background and predictability evaluation methodology, while the computational results and their discussion are given in Sect. 4. The conclusions are reported in Sect. 5. In the Appendixes the details and methods of the analytical calculations are shown.





## 2 The data

Two classes of time series are used: the observed sea surface horizontal HFR current data and the forecasted wind data in the Gulf of Trieste (Northern Adriatic Sea, Fig. 1a). The data sets used are identical to those presented in Flora et al. (2023) and

refer to a selected grid point in the Gulf of Trieste (point P and point WRF$_P$ in Fig. 1b) and cover a time range of almost two years: from January 1, 2021 to October 18, 2022.

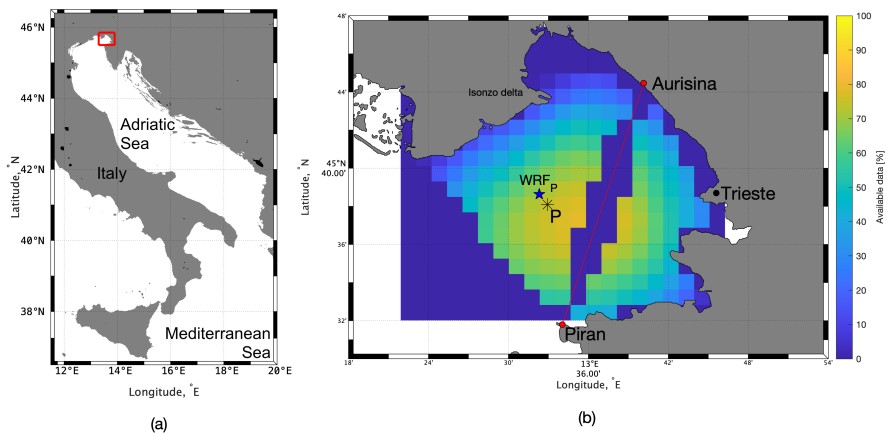

**Figure 1.** (a) Gulf of Trieste location (red rectangular) in the Adriatic Sea; (b) Gulf of Trieste zoom with percentage of available HFR data (in multiple colors) in the selected period. The HFR baseline is shown with the red line between Aurisina and Piran. The HFR "P" grid point is shown with the black asterix and the closest WRF grid point is marked with the blue star and called "WRF$_P$" (Flora et al., 2023).

Sea surface currents are measured by two beamforming WEllen RAdar stations operating in the Gulf of Trieste (Lorente et al., 2022) (Fig. 1b, red dots). The HFR system combines the radial sea surface currents of the two stations to obtain the longitudinal $u$ and the latitudinal $v$ components, with a space resolution of 1.5 km and a time resolution of 30 min. The data are open access,

further details can be found in OGS et al. (2023). The quality control standards from the EU high-frequency node (Corgnati et al., 2018) were applied to the data set. In addition, any remaining spikes were removed. The measured $u$ and $v$ time series used in this article are from the P grid point (Fig. 1b).

The atmosphere data consist of the forecasted wind velocity field at 10 m above the surface in the Gulf of Trieste from the Weather Research and Forecasting (WRF) model (https://www.mmm.ucar.edu/models/wrf), version 4.2.1. The forecasting is

performed daily by the Agenzia Regionale per la Protezione dell'Ambiente del Friuli Venezia Giulia (ARPA FVG) using initial and boundary conditions from the National Oceanic and Atmospheric Administration Global Forecasting System (https://www.ncei.noaa.gov/products/weather-climate-models/global-forecast) and provides the wind time series for the same day





and the following one. The field has a spatial resolution of 2 km and a temporal resolution of 1 h. Additional technical details can be found in Goglio (2018). The wind components time series used in this article are from the $\mathrm{WRF_P}$ grid point (Fig. 1b)

and account for the one-day forecasting.

## 3 Methods

This Section provides the model definition, the FRR background with details on how it is dealt within this study and some methodology to evaluate the predictability of the developed model.

### 3.1 The model hierarchy

The aim of the present work is to numerically simulate observed sea surface currents with a slow dynamics, that is governed by the slow components of the forcings and with fast variations that parameterise the remaining, unresolved, processes. This is achieved when the time averaged dynamics as well as the statistics of the time increments of the model agrees with the observation. In the following, the time series prefix "$\delta$" gives the increment time-series, $\delta u(t) = u(t+\delta) - u(t)$, and we call the observed fat-tailed PDF of the sea surface current increments, from Flora et al. (2023), as the "Exp-Lin" PDF, since it is

composed of an exponential and a linear term.

When modelling a natural process, as the currents, through a SDE, part of the dynamics is resolved and part of it is parameterized by noise. What is deterministic and what is noise depends on the degree of coarse-graining. In our previous work (Flora et al., 2023) all the dynamics was paramerterized and the model has no predictability. The other extreme is a purely deterministic model that includes all the processes involved. Such model is unattainable. We implemented a hierarchy of three

idealised local models (two velocity components at one point in space, evolving in time) for the sea surface current in the Gulf of Trieste. The first is the purely DETerministic (DET) model, taking into account the tidal signal and the wind-forced Ekman dynamics, while the unresolved processes are neglected. The second is the GAUssian (GAU) model, that adds Gaussian additive stochastic noise for which the corresponding SDE can be solved analytically. The third is the SUPerstatistical (SUP) model that considers a deterministic signal with the most realistic stochastic noise. The stochastic part of this model is partially resolvable

analytically. Furthermore, these models are forced by nine different Forcing Protocols (FPs) (see Sect. 4). The deterministic and stochastic models come in pairs, the more processes and time-scales are resolved deterministically, the less have to be parameterised stochastically.

More precisely, the total sea surface current vector $\boldsymbol{u_o}$ is given in terms of the tidal current $\boldsymbol{u_M}$, the wind-forced Ekman

sea surface current $\boldsymbol{u_E}$ and the stochastic current $\boldsymbol{u_S}$, as detailed in Table 1 for the different models. The tidal signal is an analytic and linear combination of $n$ tidal components $u_M(t) = \sum_{i=1}^{n} a_{u,i} \cos(\omega_i t - \phi_{u,i})$ (analogously for $v_M$), where $a$, $\omega$ and $\phi$ are the amplitude, the frequency and the phase of each tidal component, respectively.

The Ekman current, subject to the wind forcing, the Coriolis force and the friction with the underlying water-masses, is mod-





elled by:

$$
\frac{d}{dt}\begin{pmatrix} u_E \\ v_E \end{pmatrix} = \begin{pmatrix} -\frac{C_B}{\tilde{h}}\tilde{u} & f \\ -f & -\frac{C_B}{\tilde{h}}\tilde{u} \end{pmatrix}\begin{pmatrix} u_E \\ v_E \end{pmatrix} + \frac{1}{\tilde{h}}\begin{pmatrix} F_u \\ F_v \end{pmatrix}.
\tag{1}
$$

In Eq. (1), $C_B = \rho_o c_b$ where $c_b$ is the underlying ocean layer drag coefficient, $\tilde{h} = \rho_o h$ where $\rho_o$ is the ocean density and $h$ is the considered ocean surface layer depth, $\tilde{u}$ is the total speed of the surface layer as defined in Table 1. The last does not include the tidal current, since it is assumed that the tidal signal is constant along the water column and does not create shear, while the Ekman and stochastic signals are present in the surface layer only. Finally $f$ is the Coriolis parameter and $\boldsymbol{F} = \rho_a c_a |\boldsymbol{u_a} - \boldsymbol{u_o}|(\boldsymbol{u_a} - \boldsymbol{u_o})$ is the wind forcing ($\rho_a$ is the atmosphere density, $c_a$ the atmosphere drag coefficient and $\boldsymbol{u_a}$ the wind velocity at 10 m above the sea surface). Eq. (1) is an ordinary differential equation of the DET model, while it includes the stochastic quantity $\tilde{u}$ in the GAU and SUP models. If the wind forcing were to cease, the modelled mean flow would fall to the tidal signal.

The Ekman modelization in Eq. (1) is only valid on frequency scales comparable to the Coriolis parameter. The stochastic velocity, representing the fast unresolved turbulent dynamics and present in the GAU and SUP models, is defined by the following set of equations:

$$
d\begin{pmatrix} x \\ y \end{pmatrix} = -\gamma_x \begin{pmatrix} x \\ y \end{pmatrix} dt + \begin{pmatrix} \sqrt{Q_u} & 0 \\ 0 & \sqrt{Q_v} \end{pmatrix}\begin{pmatrix} dW_x \\ dW_y \end{pmatrix}
\tag{2}
$$

$$
\frac{d}{dt}\begin{pmatrix} u_S \\ v_S \end{pmatrix} = -\gamma_u \begin{pmatrix} u_S \\ v_S \end{pmatrix} + \eta \begin{pmatrix} x \\ y \end{pmatrix}
\tag{3}
$$

In Eq. (2) and Eq. (3) the variables $dW_x$ and $dW_y$ are derived from independent Wiener stochastic processes. For every independent realization of the Wiener processes $W_x$ and $W_y$, we obtain a solution of the dynamics, named a random walker. The collection of all the random walkers forms the stochastic ensemble and allows to construct a PDF. The coefficient $\gamma_x = 1/\tau$ is the inverse of the characteristic short time scale $\tau$ (short compared to the long time scale discussed below in connection with the SUP model) whose numerical value is determined from the observations (Flora et al., 2023), $\gamma_u$ is a coefficient which is adjusted to obtain the observed decay of the autocorrelation function of the SUP model sea surface currents and $\eta$ is a proper constant discussed below. The difference between the GAU and the SUP model lies in the terms $Q$. In the GAU model, that is a simplification of the SUP model, $Q_u$ and $Q_v$ are constants: they are the mean values of the SUP model $Q$ variables as given in Table 1. The GAU model is a system of linear SDEs and all variables are Gaussian and are determined by their means and variances. All the dependencies for means and variances on the original parameters are given in Appendix A. Furthermore in Eq. (A32) the analytical condition on $\eta$ for which the Gaussian variable $\delta u_S$ has the same mean and variance as $x$ (analogously $\delta v_S$ with $y$), given a fixed $\gamma_u$, is shown. This also helps to adjust the parameters of the SUP model.



In the SUP model, the $Q$ terms originate themselves from stochastic processes $\alpha_i$ governed by:

$$d\alpha_i = -\mu\alpha_i dt + \beta dW_i, \qquad i = 1, \cdots, 2\nu \qquad \text{and} \qquad \beta = \begin{cases} \beta_u & \text{for the } u \text{ component} \\ \beta_v & \text{for the } v \text{ component} \end{cases} \tag{4}$$

where the variables $dW_i$ are derived from independent Wiener stochastic processes, $\beta_u$ and $\beta_v$ are constants whose values are determined empirically, $\mu = 1/T$ is the inverse of the characteristic long time scale $T$ (i.e. $\tau \ll T$), corresponding to the variation time-scale of the variance in the superstatistical approach. Its numerical value is determined from the observations (Flora et al., 2023). The variables $\alpha_i$ are solutions of the Langevin equation: they are Ornstein-Uhlenbeck processes, characterized by a Gaussian statistics:

$$p_\alpha(\alpha_i) = \sqrt{\frac{\mu}{\pi\beta^2}} e^{-\mu\alpha_i^2/\beta^2}. \tag{5}$$

In the SUP model the variables $Q = \sum_{i=0}^{2\nu} \alpha_i^2$ depend on the variables $\nu$ and $\alpha_i$ (see Eq. 4), as shown in Table 1. The variable $\nu$ gives the degrees of freedom (dof), according to the interpretation of Flora et al. (2023): the system has one dof if its positive variable characterizing the variability of the system maximazes the entropy, i.e. it is exponentially distributed. The $Q$ variables are distributed according to a Gamma distribution with shape parameter equal to $\nu$, given in Eq. (B5) and Table B1. In Eq. (B6) we give the computation for $\langle Q \rangle$ used in the GAU model. In Appendix B the PDFs of the variable $x$ (and $y$) are derived for $\nu = 2$ and $\nu = 1/2$. In the case of 2 dof, the variables $x$ and $y$ are Exp-Lin distributed with a PDF given by:

$$\begin{cases} p_{\nu=2}(x) = \frac{\sqrt{\mu\gamma_x}}{2\beta_u} e^{-2\sqrt{\mu\gamma_x}|x|/\beta_u} \left( \frac{2\sqrt{\mu\gamma_x}}{\beta_u}|x| + 1 \right) \\ p_{\nu=2}(y) = \frac{\sqrt{\mu\gamma_x}}{2\beta_v} e^{-2\sqrt{\mu\gamma_x}|y|/\beta_v} \left( \frac{2\sqrt{\mu\gamma_x}}{\beta_v}|y| + 1 \right). \end{cases} \tag{6}$$

In the case of 1/2 dof, the variables $x$ and $y$ are distributed according to a modified Bessel function of the second kind of zero order $K_0(|z|)$:

$$\begin{cases} p_{\nu=\frac{1}{2}}(x) = \frac{2\sqrt{\mu\gamma_x}}{\pi\beta_u} K_0 \left( \frac{2\sqrt{\mu\gamma_x}}{\beta_u}|x| \right) \\ p_{\nu=\frac{1}{2}}(y) = \frac{2\sqrt{\mu\gamma_x}}{\pi\beta_v} K_0 \left( \frac{2\sqrt{\mu\gamma_x}}{\beta_v}|y| \right). \end{cases} \tag{7}$$

The SDEs for the variables $u_S$ and $v_S$ in Eq. (3) contain a linear drag term and are forced by a coloured non-Gaussian noise. We do not know an analytical distribution of $u_S$ and $v_S$. Nevertheless, thanks to the definition of the constant $\eta$ in Eq. (A32) discussed in connection with the GAU model, the distributions of $\delta u_S$ and $\delta v_S$ are numerically similar to the PDFs of $x$ and $y$ (shown in Sect. 4.1).

As it can be seen in Eq. (6) and Eq. (7), the final stochastic PDFs depend on the $\beta$ parameter, that is linked to the variable's second order moment, by $E[x^2] = \frac{\beta^2}{\mu\gamma_x}$ for $\nu = 2$. For this case, it will be shown in Sect. 4.1 that the PDF of the total velocity increment is given mainly by the stochastic velocity increment PDF, only. For this reason in the numerical simulations with 2 dof in the stochastic part of the model the value of $\beta$ is therefore fixed by the observed variance $s^2$ of the HFR velocity increment found in Flora et al. (2023): $\beta = s\sqrt{\mu\gamma_x}$. In the case of 1/2 dof in the stochastic signal, the tidal and Ekman signal





contribute more to the variance of the total velocity increment, but their contribution is unknow analytically. For this reason in this case the $\beta$ coefficient is increased empirically.

In Table 1 the definition of the variables distinguishing the DET, GAU and SUP models is given with a summary of the main

| | $\boldsymbol{u_o}$ | $\tilde{u}$ | $Q$ | $x$ and $y$ distribution | $\delta u_S$ and $\delta v_S$ distribution |
|---|---|---|---|---|---|
| DET | $\boldsymbol{u_E} + \boldsymbol{u_M}$ | $|\boldsymbol{u_E}|$ | no stochasticity | no stochasticity | no stochasticity |
| GAU | $\boldsymbol{u_E} + \boldsymbol{u_M} + \boldsymbol{u_S}$ | $|\boldsymbol{u_E} + \boldsymbol{u_S}|$ | $\langle Q \rangle_{SUP} = \nu\beta^2/\mu$ | Gaussian, Eq. (A28) | Gaussian, Eq. (A30) |
| SUP | $\boldsymbol{u_E} + \boldsymbol{u_M} + \boldsymbol{u_S}$ | $|\boldsymbol{u_E} + \boldsymbol{u_S}|$ | $\sum_{i=1}^{2\nu} \alpha_i^2$ | $\nu = 2$: Exp-Lin, Eq. (6) $\nu = 1/2$: Bessel, Eq. (7) | not analytical, but numerically similar to the $x$ and $y$ PDF |

**Table 1.** Variables definition distinguishing the different models and $x$, $y$, $\delta u_S$ and $\delta v_S$ PDFs' characteristics. The variable $\nu$ indicates the degrees of freedom of the stochastic signal.

properties of the models' variables. For all the models we can consider, or not, the eddy depletion term in the wind stress term, i.e. the relative velocity of the wind with respect to the sea surface current, causing a reduction of kinetic energy injection into

the ocean (Zhai et al., 2012), in contrast to the absolute wind velocity. We report the results just with the eddy depletion term, since it is the theoretically most correct formulation and it does not significantly burden the computational calculation.

In summary, we developed three types of models: the DET model is purely deterministic with tidal and Ekman currents, the GAU model adds a Gaussian stochastic signal, while the SUP model considers a stochastic sea surface current with fat-tailed increments instead of the Gaussian noise (Table 1).

**3.2 The FRR background and methods**

The FRR relates the response of a system pertubed by an external perturbation to internal fluctuations of the unperturbed system. Exploring the FRR is particularly challenging and interesting when considering natural systems where perturbation experiments cannot be performed and statistical ensembles are not available. Our SUP model is an example of a subcomponent of the climate system and it is instructive to explore the FRR for such model. We refer to Lacorata and Vulpiani (2007) for the

theoretical background concerning FRRs. We show here some foundamentals of the theoretical concepts and the methodology we applied to the most involved of the SUP models (FP9, Table 2, Sect. 4).

In the present case, given the SUP model evolutionary system with the SUP variables $u_o = u_E + u_M + u_S$ and $v_o = v_E + v_M + v_S$, a small fixed perturbation $\Delta u_S$ or either $\Delta v_S$ is imposed at $t = t_0$ to each walker. Then the system is left free to

evolve and, for each time, the separation $\Delta u_o(t|t_0) = u_{o,P}(t) - u_o(t)$ and $\Delta v_o(t|t_0) = v_{o,P}(t) - v_o(t)$ between the perturbed ($u_{o,P}$ and $v_{o,P}$) and the unperturbed ($u_o$ and $v_o$) systems is computed. We emphasise that the methodology requires one velocity component to be perturbed and not both in the same perturbed simulation. It is possible to obtain the mean value of the





perturbed variables through a mean response function $R_{uu}(t)$ or $R_{vv}(t)$:

$$\langle \Delta u_o(t) \rangle = R_{uu}(t) \Delta u_S \qquad \text{when } u_S \text{ is perturbed}$$

$$\langle \Delta v_o(t) \rangle = R_{vv}(t) \Delta v_S \qquad \text{when } v_S \text{ is perturbed} \tag{8}$$

where the angle brackets $\langle \cdots \rangle$ indicate the ensemble mean. The FRR is concerned with expressing the mean response function in terms of correlation functions of the unperturbed system.

In practice, the mean response functions is computed as follows. After the first perturbation at $t_0$, again at $t_k = t_0 + k\Delta t$ the selected variable, in the perturbed system, is perturbed with the same variation and the separation between the perturbed and the unperturbed systems is computed. The procedure is repeated $M = 1311 \gg 1$ times with $\Delta t = 12$ h, $\Delta u_S = \Delta v_S = 8$ cm/s.

The diagonal mean response functions are computed as follows:

$$R_{uu}(t) = \frac{1}{M} \sum_{k=0}^{M-1} \frac{\langle \Delta u_o(t_k + t | t_k) \rangle}{\Delta u_S} \qquad \text{when } u_S \text{ is perturbed}$$

$$R_{vv}(t) = \frac{1}{M} \sum_{k=0}^{M-1} \frac{\langle \Delta v_o(t_k + t | t_k) \rangle}{\Delta v_S} \qquad \text{when } v_S \text{ is perturbed.} \tag{9}$$

According to Lacorata and Vulpiani (2007), the FRR holds if the mean response functions have a connection with some suitable correlation functions computed in the unperturbed system. In particular, in the case of multivariate Gaussian variables, the mean response functions are a linear combination of the variables correlations. In Sect. 4.2 the results with unperturbed

initial condition $\boldsymbol{u_o}(t = 0) = 0$ and when the stochastic signal is initially perturbed, as described before, is shown. Some brief comments for the case when the Ekman system is initially perturbed are also provided.

### 3.3 Predictability evaluation methods

In order to test predictability capabilities of the SUP model, data assimilation methods are adopted with the following assumption: the HFR observations are not affected by observational uncertainty and represent the reality. Every $\Delta t$ time the perturbed

system $(\boldsymbol{u_{o,P}})$ is updated to the observed HFR velocities (both the $u_o$ and $v_o$ components in the same simulation), in particular it is the stochastic signal to be perturbed. The method may appear equivalent to the FRR method, but the equivalence is not valid, since: (i) the initial perturbation is not a constant, but changes for each $\Delta t$ time window and for each walker; (ii) we are perturbing both the $u_o$ and $v_o$ components (in the stochastic signal) in the same perturbed simulation. We define the following functions:

$$\xi(t) = \sqrt{\frac{\frac{1}{M}\sum_{k=0}^{M-1}\langle(\boldsymbol{u_{o,P}}(t_k+t) - \boldsymbol{u_o}(t_k+t))^2\rangle}{\frac{1}{M}\sum_{k=0}^{M-1}\langle(\boldsymbol{u_{o,P}}(t_{k-1}+\Delta t) - \boldsymbol{u_{HFR}}(t_k))^2\rangle}} \tag{10}$$

$$\epsilon(t) = \sqrt{\frac{\frac{1}{M}\sum_{k=0}^{M-1}\langle(\boldsymbol{u_{o,P}}(t_k+t) - \boldsymbol{u_{HFR}}(t_k+t))^2\rangle}{\frac{1}{M}\sum_{k=0}^{M-1}\langle(\boldsymbol{u_{o,P}}(t_{k-1}+\Delta t) - \boldsymbol{u_{HFR}}(t_k))^2\rangle}} \tag{11}$$





where the time $t_{k-1} + \Delta t$ means $t_k$ right before the perturbation. The function $\xi(t)$ quantifies the difference between the perturbed and the unperturbed system, while the function $\epsilon(t)$ computes the difference between the perturbed system and the observations. Both are normalized through the mean initial perturbation over all the time windows. In Sect. 4.3 the results are presented. Some brief comments for the case when the Ekman system is initially perturbed are also provided.

## 4 Results and discussion

We test the different models imposing different Forcing Protocols (FPs) in the tidal and wind signal (Table 2). In the most complete FP the complete tidal and wind forcing is included (FP9), in the other FPs the forcing signal is either averaged by a moving average over 12 h or completely omitted.

| Forcing Protocol | | 1 | 2 | 3 (n.s.) | 4 (n.s.) | 5 | 6 (n.s.) | 7 (n.s.) | 8 | 9 |
|---|---|---|---|---|---|---|---|---|---|---|
| Forcings | tide | ✗ | ✗ | ✗ | $\overline{12h}$ | $\overline{12h}$ | $\overline{12h}$ | ✓ | ✓ | ✓ |
| | wind | ✗ | $\overline{12h}$ | ✓ | ✗ | $\overline{12h}$ | ✓ | ✗ | $\overline{12h}$ | ✓ |
| Results | dof $\nu$ | 2 | 2 | 2 | 2 | 2 | 2 | 1/2 | 1/2 | 1/2 |
| | $\frac{\beta}{s\sqrt{\mu\gamma_x}}$ | 1 | 1 | 1 | 1 | 1 | 1 | 1.5 | 1.5 | 1.5 |

**Table 2.** Settings and results of the experiments on the models: the ✗ symbol means that the forcing is set to 0, the $\overline{12h}$ symbol means that the forcing is filtered with a 12 h moving average, the ✓ symbol means that we use the original forcing. The variable $s$ is the observed standard deviation of the HFR velocity increment found in Flora et al. (2023). The abbreviation n.s. stays for "not shown".

The numerical values of the coefficients common to all the models and to all the runs are described here. Tidal harmonic analysis calculations on the HFR sea surface currents were done using the MatLab programme t_tide (Pawlowicz et al., 2002). The resultant tidal ellipse parameters are shown in Table 3. The $S1$, $M2$ and $S2$ tidal components show a sufficiently high Signal-to-Noise Ratio (SNR) and are therefore included to be considered in our modelization. The conversion from the ellipse parameters to the sinusoidal tidal coefficients is applied according to Foreman (1978). The non-filtered tidal signal consists of the $S1$, $M2$ and $S2$ components, while the 12 h moving averaged signal is characterized by the diurnal oscillation only. The wind velocity $\boldsymbol{u_a}$ time series is linearly interpolated from its 1 h original resolution or from a 12 h moving average to the

| tide | $\omega$ [h$^{-1}$] | SEMA [cm s$^{-1}$] | SEMI [cm s$^{-1}$] | INC [°] | PHA [°] | SNR |
|---|---|---|---|---|---|---|
| $S1$ | 0.0426667 | $6.732 \pm 0.692$ | $-2.522 \pm 0.66$ | $-26.46 \pm 6.65$ | $233.62 \pm 7.64$ | 95 |
| $M2$ | 0.0805114 | $3.537 \pm 0.407$ | $0.296 \pm 0.39$ | $47.84 \pm 6.05$ | $151.27 \pm 5.96$ | 76 |
| $S2$ | 0.0833333 | $3.389 \pm 0.454$ | $-0.167 \pm 0.39$ | $23.67 \pm 7.22$ | $130.69 \pm 7.59$ | 56 |

**Table 3.** Tidal ellipse parameters with 95 % confidence interval estimates, extracted from the HFR data.

integration time step of the numerical model (2.5 min). The determination of $\tilde{h}$ and $C_B$ as best linear fit functions of the wind speed is given in Appendix C. The algorithm to determine the wind regimes Bora, Sirocco, Mistral and Low wind is explained



in Flora et al. (2023): if the daily wind speed is higher than the threshold value of 3 m/s, then it is defined as Bora, Sirocco or Mistral depending on its direction; if it is lower, it is classified as Low wind. Mistral occures just for a few days in almost two years, it is therefore considered statistically not significant. In the numerical integration the parameters shown in Table 4 are used and a Runge-Kutta method of the second order is adopted for the Ekman system.

| parameter | value |
|---|---|
| total integration time | 656 days |
| Runge Kutta time step | $3\times10^2$ s |
| $\delta$ | $1.44\times10^4$ s |
| $\rho_a$ | 1.3 kg m$^{-3}$ |
| $c_a$ | $10^{-3}$ |
| ensemble size | $10^5$ |
| stochastic time step $dt$ | $1.5\times10^2$ s |
| $T$ | $1.746\times10^5$ s |
| $\tau$ | $6.6\times10^3$ s |
| $\mu = 1/T$ | $5.728\times10^{-6}$ s$^{-1}$ |
| $\gamma_x = 1/\tau$ | $1.515\times10^{-4}$ s$^{-1}$ |
| $s_u$ | $12.265\times10^{-2}$ m s$^{-1}$ |
| $s_v$ | $9.6696\times10^{-2}$ m s$^{-1}$ |
| $s_u\sqrt{\mu\gamma_x}$ | $3.612\times10^{-6}$ m s$^{-2}$ |
| $s_v\sqrt{\mu\gamma_x}$ | $2.848\times10^{-6}$ m s$^{-2}$ |
| $\gamma_u$ | $1.166\times10^{-4}$ s$^{-1}$ |
| $\eta$ | $1.660\times10^{-4}$ s$^{-1}$ |

**Table 4.** Involved parameters in the numerical modelization. The numerical values of $\delta$, $\tau$, $T$, $s_u$ and $s_v$ are taken from (Flora et al., 2023).

### 4.1 Comparison of the DET, GAU and SUP models on different focings time scales

All the models start from the initial condition $\boldsymbol{u}_o(t=0) = \boldsymbol{u}_{\text{HFR}}(t=0)$, where $\boldsymbol{u}_{\text{HFR}}$ is the observed HFR sea surface velocity, with $\boldsymbol{u}_{\boldsymbol{S}}(t=0) = 0$. The obtained numerical sea surface currents are then compared to the HFR measurements. We start this section commenting FP1 which has no tides and no wind forcing (see Table 2) and compare the DET, GAU and SUP models in the FP5 and FP9 cases (see Table 2). We conclude this section with a discussion on the change of the stochastic dof when the tidal forcing is modified. We only show time series for the $u$-component of the sea surface current variables, results for the $v$-component are similar.

The stochastic model has to be adapted to the FPs to best represent the dynamics unresolved by the deterministic part of





the model. The FP1 does not take into account any physical forcings and the deterministic part vanishes. The DET model is therefore trivial, while the GAU and SUP models have a non-trivial stochatic part. The SUP model best fits the observed data when $\nu = 2$ dof are chosen (see Flora et al. (2023)), as we obtain a total velocity increment ($\delta u_o = \delta u_S$, because in FP1 $\delta u_M = \delta u_E = 0$) distributed according to the Exp-Lin PDF (as summarized in Table 1). The result is shown in Fig. 2 left column, in which SUP $\delta u_o$ of FP1 coincides with SUP $\delta u_S$ of FP5 (as shown in Table 2 and commented below, they both have $\nu = 2$ stochastic dof). The GAU model in FP1 (Eq. (A30) and Fig. 2 left column, in which GAU $\delta u_o$ of FP1 coincides with GAU $\delta u_S$ of FP5) fails in reproducing the fat-tailed Exp-Lin PDF.

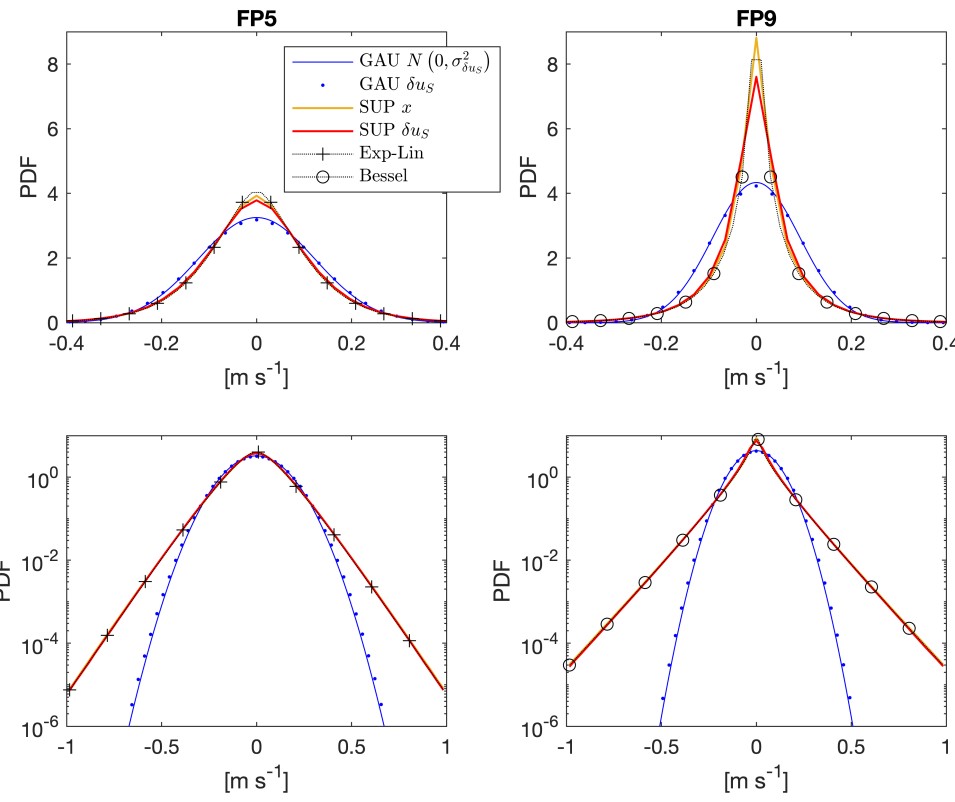

**Figure 2.** PDFs of the stochastic variables of the GAU and SUP models for FP5 (left column) and FP9 (right column) in lin-lin plots (top row) and lin-log plots (bottom row). In particular, analytical solution of the GAU $\delta u_S$ variable from Eq. (A30) (blue continuous line) and PDFs of the models variables: numerical GAU $\delta u_S$ (blue dots), SUP $x$ (yellow continuous line), SUP $\delta u_S$ (red continuous line). PDFs of the analytical superstatistical Exp-Lin PDF from Eq. (6) (black dotted line with crosses) and of the analytical superstatistical Bessel function from Eq. (7) (black dotted line with circles) are also shown. Note that in FP5 (FP9) the SUP $x$, the SUP $\delta u_S$ and the Exp-Lin (Bessel) lines almost superpose.





In the FP5 a 12 h moving-averaged tidal and wind forcing is applied to the models, while the FP9 includes the forcings without averaging (see Table 2). For the different types of forcings we choose the dof that best fitted the superstatistical Exp-Lin PDF from Flora et al. (2023). In particular, for FP5 $\nu = 2$, while for FP9 $\nu = 1/2$ give the best fits. In Fig. 2 the PDFs of the stochastic variables of the GAU and SUP models for FP5 and FP9 are compared to the analytical PDFs. The figures

validate the choice of the parameters in our model hierarchy for the different forcings, as summarized in Table 1: (i) the GAU $\delta u_S$ follows the analytical Gaussian from Eq. (A30); (ii) in FP5 the stochastic variable SUP $x$ is Exp-Lin distributed while in FP9 it is Bessel distributed; (iii) the SUP $\delta u_S$ are distributed as SUP $x$ in both the simulations, showing the validity of the choice of $\eta$ (fixed according to the analytical condition from the GAU model in Eq. (A32), its choice is discussed in Sect. 3.1 in connection with the GAU and SUP models). This also validates the numerical model through the cases where the ananalytical

solution is known.



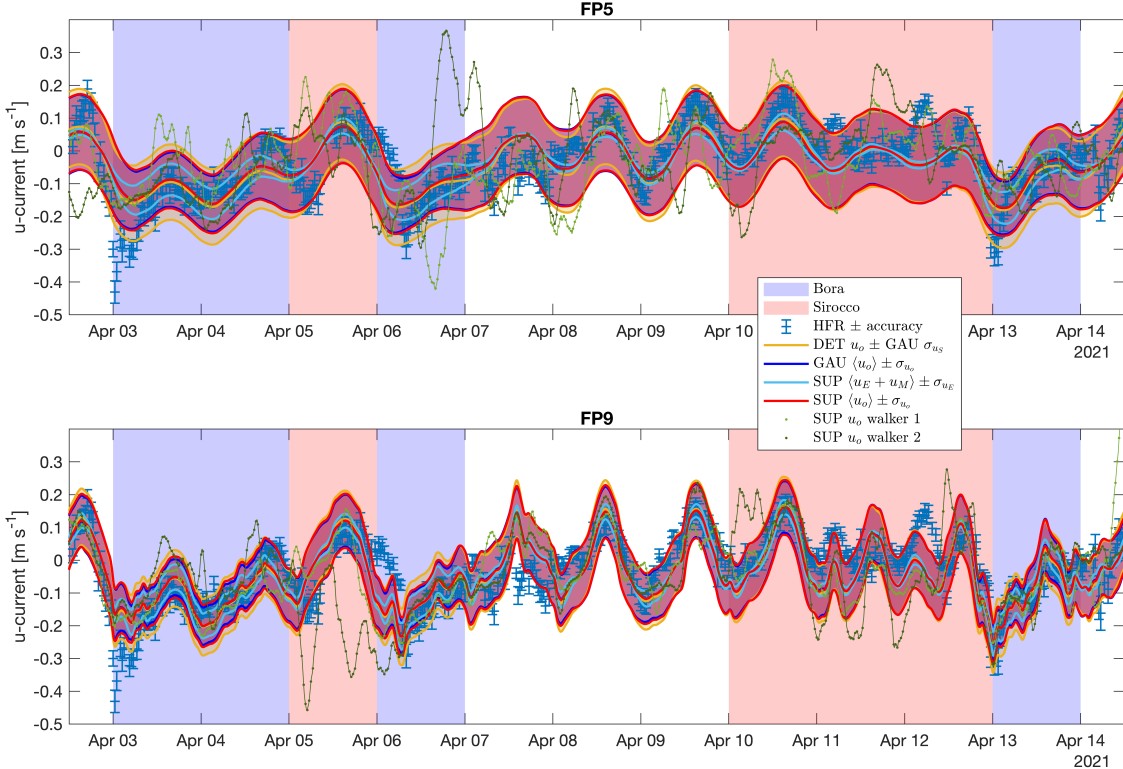

**Figure 3.** Sea surface currents time series ($u$-component) during eleven days of April 2021 of the HFR observations with their measure accuracy (blue sticks with error bars) and from the models for FP5 (top) and for FP9 (bottom): DET model $u_o$ with the Gaussian GAU model ensemble standard deviation of the stochastic velocity (yellow lines), GAU model ensemble mean of $u_o$ with their ensemble standard deviation (blue lines), SUP model ensemble mean of $u_E + u_M$ with their ensemble standard deviation (light blue lines), SUP model ensemble mean of $u_o$ with their ensemble standard deviation (red lines). The SUP $u_o$ path of two random walkers is reported (light and dark green dotted lines). The symbol $\langle \cdots \rangle$ in the legend stands for the ensemble mean. The blue and red shadings represent Bora and Sirocco wind regimes respectively, seen in Flora et al. (2023).





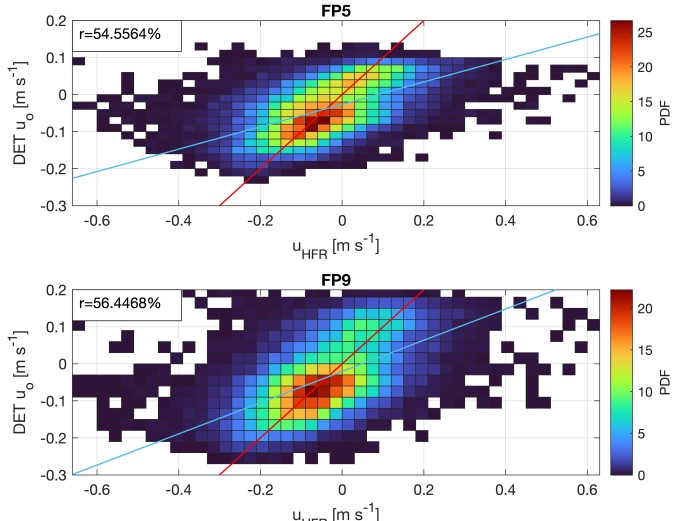

**Figure 4.** Normalised 2D histograms of the correlation between the HFR observations and the DET model $u$-component velocities from the simulated period (almost two years), from FP5 (top) and from FP9 (bottom). The $r$ correlation coefficient is reported. The light blue line is the linear regression of the scattered data, while the red line is the bisector.

In Fig. 3 the time series of the observed and model velocities for some days of April 2021 are reported. The difference in the forcing time scale is clearly visible in the deterministic DET model velocities variability: smoother in FP5 than in FP9. They both follow the slow variability of the observed HFR currrents. In fact, looking at Fig. 4, the linear regression of the scattered data has a lower slope with respect to the bisector, meaning that the DET model is not able to simulate the observed extremes. Nevertheless, their 2D histograms have an elongated peak along the bisector and their correlation coefficient reaches 55 % in FP5 and 56 % in FP9 for the $u$-component (29 % and 35 % for the $v$-component, respectively). Moreover, it is possible to see that sometimes (e.g. 3rd of April 2021 from Fig. 3) the observed $u_{\text{HFR}}$ sea surface current shows a negative peak at the beginning of Bora events. This behaviour is missing in the DET model $u_o$ velocity. This fact can be due to the model istantaneous change of the values of $\tilde{h}$ and $C_B$ with the wind speed, while the sea surface system in nature takes some time to adjust to the wind forcing through vertical mixing (the thin surface layer suddenly accelerates, then thickens and the speed decreases). Regarding the ensemble averaged sea surface currents of the GAU and SUP models, they follow closely the DET model time series. This shows that the stochastic models increase the variability to observed values, but have little influence on the averaged dynamics. The SUP ensemble standard deviation of the $\boldsymbol{u_E} + \boldsymbol{u_M}$ velocity (light blue band) is smaller than the SUP total velocity ensemble standard deviation $\sigma_{u_o}$ (red band), as expected, since the tidal velocity is purely deterministic and in the Ekman velocity the stochasticity enters only through the drag. What is particularly interesting is that the GAU $\sigma_{u_o}$ ensemble standard deviation (blue band) almost completely coincides with the SUP $\sigma_{u_o}$ (red band) and not with the GAU $\sigma_{u_S}$ (yellow band) ensemble standard deviation. A clear example of that is visible on the 3rd, 4th, 6th and 13th of April 2021 in Fig. 3 for the $u$-component, where the wind forcing increases and the GAU and SUP $\sigma_{u_o}$ standard deviations (blue and red bands respec-



tively) reduce with respect to the GAU $\sigma_{u_S}$ statistics (yellow band). In this case it originates from the stochasticity present in

the Ekman drag term.

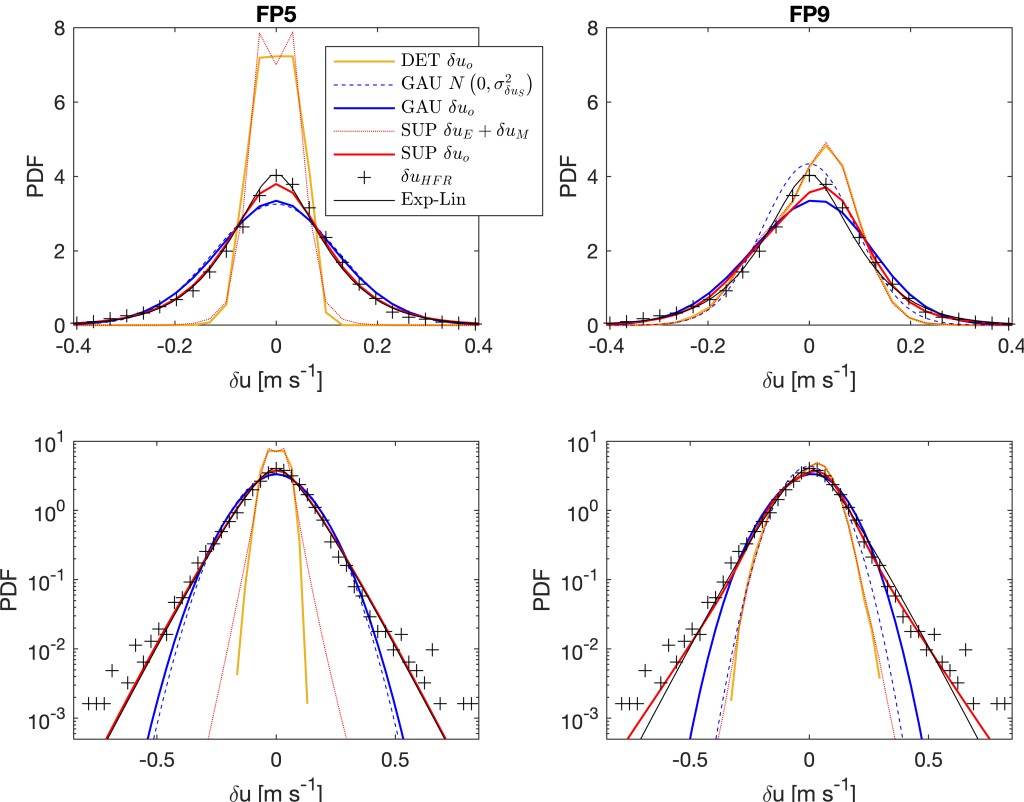

**Figure 5.** PDFs of the observed, analytical and numerical velocity-increments from FP5 (left column) and FP9 (right column) in lin-lin plots (top row) and lin-log plots (bottom row). In particular, we show the PDFs of the observed HFR velocity-increment (black crosses), of the analytical superstatistical PDF from Eq. (6) (black line), of the analytical Gaussian of the GAU $\delta u_S$ variable from Eq. (A30) (blue dashed line) and of the models variables: DET $\delta u_o$ (yellow line), GAU $\delta u_o$ (blue line), SUP $\delta u_E + \delta u_M$ (dotted red line), SUP $\delta u_o$ (full red line).

Looking at the PDFs of the velocity increments in Fig. 5, it can be seen that the DET model PDF in the FP5 is more peaked around zero, as compared to FP9 which is closer to the observations. This is because the latter model has a higher variability due to the inclusion of higher frequency forcing data. Despite the rather high correlation of the DET currents with the observations,

the DET model does not explain most of the variability seen in the HFR data, especially the tails, the extreme events, are not well reproduced.

From Fig. 5 one can also observe that in FP5 the GAU $\delta u_o$ PDF is similar to the analytical PDF $N(0, \sigma_{\delta u_S})$ from Eq. (A30), while in FP9 it deviates, showing a lower peak and fat tails. The reason of that is explained in more detail for the SUP model





in the next paragraph, but in summary it is due to the capacity in the FP9 of the tidal and Ekman modelization to resolve large
part of the variability. For both the FPs, although the first and second moments of the velocities are comparable to the SUP
model (Fig. 3) and the shape of the PDF of the velocity increments has improved as compared to the results of the DET model
(Fig. 5), the tails of the GAU velocity increment PDFs are (by construction) not fat enough to resemble the superstatistical
Exp-Lin PDF and the occurance of extreme events is strongly under estimated.

Regarding the SUP model, in FP5 the SUP $\delta u_E + \delta u_M$ PDF is very peaked, due to the 12 h forcing time scale that suppresses a
large part of the variability. Hence, the SUP $\delta u_E + \delta u_M$ PDF contributes little to the SUP $\delta u_o$ PDF, that is distributed similar to
the SUP $\delta u_S$ Exp-Lin PDF, having fat tails (compare Fig. 2 and Fig. 5). In FP9 the SUP $\delta u_E + \delta u_M$ PDF is distributed closely
to the DET $\delta u_o$ PDF (larger with respect to the FP5 case), meaning that a large part of the SUP $\delta u_o$ variability is resolved by
the deterministic model. The SUP $\delta u_S$ PDF is given by the Bessel function of Eq. (7) and shown in Fig. 2. The SUP $\delta u_o$ PDF,
shown in Fig. 5, is a convolution of the SUP $\delta u_E + \delta u_M$ PDF (no analytical form available) and the Bessel function. This
PDF shows a lower and shifted peak with respect to the observed and superstatistical Exp-Lin PDFs (due probably to the SUP
$\delta u_E + \delta u_M$ PDF), but its tails deviate slightly from the observed and Exp-Lin ones.

The percentages of the observed HFR velocity increments inside the SUP $\sigma_{\delta u_o}$ (and $\sigma_{\delta v_o}$) ensemble standard deviation band
are measured from the model results:

$$\text{FP5} \begin{cases} \Pr\left(\delta u_o - \sigma_{\delta u_o} \leq \delta u_{HFR} \leq \delta u_o + \sigma_{\delta u_o}\right)^{FP5}_{SUP} \simeq 73\% \\ \Pr\left(\delta v_o - \sigma_{\delta v_o} \leq \delta v_{HFR} \leq \delta v_o + \sigma_{\delta v_o}\right)^{FP5}_{SUP} \simeq 73\% \end{cases} \tag{12}$$


$$\text{FP9} \begin{cases} \Pr\left(\delta u_o - \sigma_{\delta u_o} \leq \delta u_{HFR} \leq \delta u_o + \sigma_{\delta u_o}\right)^{FP9}_{SUP} \simeq 61\% \\ \Pr\left(\delta v_o - \sigma_{\delta v_o} \leq \delta v_{HFR} \leq \delta v_o + \sigma_{\delta v_o}\right)^{FP9}_{SUP} \simeq 62\% \end{cases} \tag{13}$$

Repeating the calculation in Eq. (12) and Eq. (13) without the eddy depletion term, we obtain a difference of less than 1 %
with respect to the shown case, in which the relative velocity between the sea and the atmosphere is considered. For FP5 the
percentages compare well to the analytical superstatistical Exp-Lin values:

$$\Pr(-\sigma \leq x \leq \sigma) = \int_{-\sigma}^{+\sigma} \frac{e^{-2|x|/\sigma}\left(\frac{2|x|}{\sigma} + 1\right)}{2\sigma} dx = \int_{0}^{+\sigma} \frac{e^{-2x/\sigma}\left(\frac{2x}{\sigma} + 1\right)}{\sigma} dx = \frac{1}{2}\int_{0}^{2} e^{-z}(z+1)dz \simeq 73\% \tag{14}$$

This fact quantitatively confirms that the SUP model in FP5 needs $\nu = 2$ dof in the stochastic signal, producing a reliable PDF
with respect to the observations. In this respect, the observations are therefore indistinguishable from a single walker of the
SUP model ensemble. The percentages of the FP9 case, that has $\nu = 1/2$ dof in the stochastic signal, show small discrepancies
with respect to the analytical ones. Despite the deviations from the analytical superstatistical Exp-Lin PDF, the SUP model
in FP9 reproduces the PDF tails of the velocity increments with even greater fidelity to the observations than in the FP5 case
(Fig. 5).



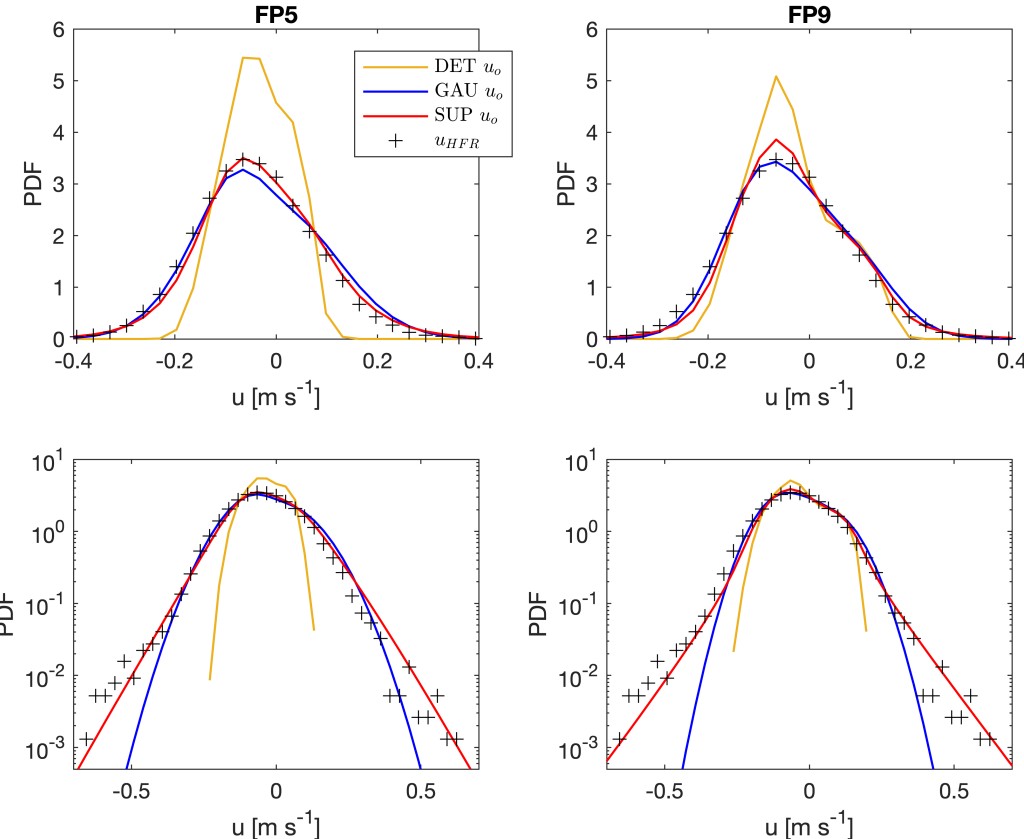

**Figure 6.** PDFs of the observed $u_{HFR}$ (black crosses) and numerical models sea surface currents from FP5 (left column) and FP9 (right column) in lin-lin plots (top row) and lin-log plots (bottom row): DET $u_o$ (yellow line), GAU $u_o$ (blue line), SUP $u_o$ (red line).

We move now our attention from the PDFs of the velocity increments to the PDFs of the velocities (Fig. 6). The DET model is not able to represent the observed PDFs, with a FP5 clearly missing the most of the variability and a FP9 (full forcing) improving the situation, but still not reproducing the fat tails. A stochastic part is needed to represent unresolved processes and

to increase the variability. The GAU and SUP models can reproduce better the observations. The GAU model shows a good pattern around the peak, while it fails in the reproduction of the extreme events, with errors of around one and three orders of magnitude for the occurance of extreme velocities of 0.5 m s$^{-1}$ in the FP5 and FP9, respectively. The SUP model in FP5 confirmes, also with this variable, to be able to catch the observed statistics. In FP9 the SUP model shows a higher peak, but demonstrates to reproduce some details that FP5 can not, as the PDF decay with two different slopes.





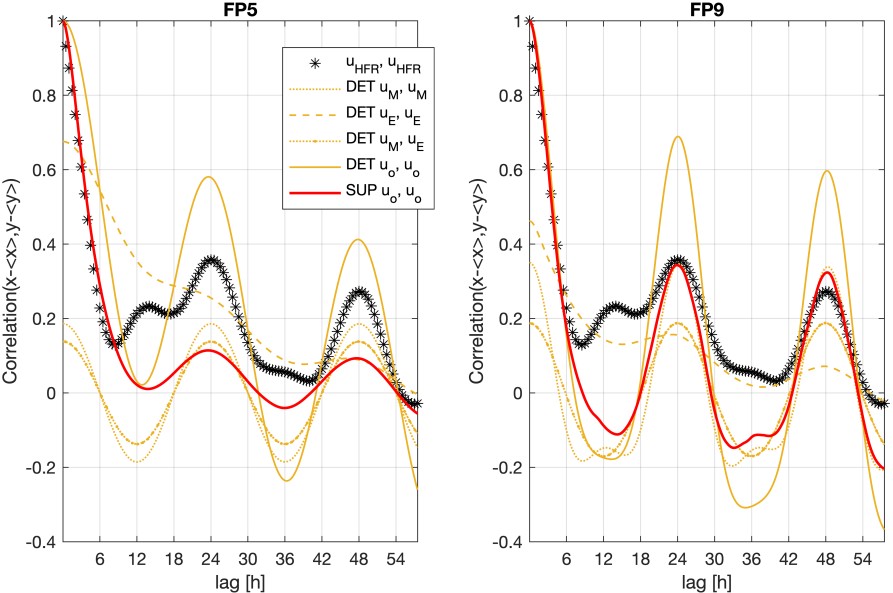

**Figure 7.** Temporal auto-correlation function of the $u$-component of the HFR sea surface velocities (black asterisks) and of the models variables from FP5 (left) and FP9 (right): auto-correlation of the DET $u_o$ (continuous yellow line) and of the SUP $u_o$ (continuous red line). The components of the DET $u_o$ auto-correlation are reported: the auto-correlation of the DET $u_M$ (dotted yellow line), of the DET $u_E$ (dashed yellow line), and the cross-correlation of the DET $u_M$ and $u_E$ (yellow dotted line with dots), where the normalization is based on the DET $u_o$ auto-correlation.

Regarding the deterministic DET model $u_o$ temporal auto-correlation functions shown in Fig. 7, the periodicity of the peaks are consistent with what is observed from the HFR data, while the amplitude of the peaks is higher with respect to the observations. This fact is mainly due to the high auto-correlation of the tidal signal and the absence of stochastic variability. The cross-correlation between the DET $u_M$ and the DET $u_E$ velocities shows a clear undamped periodicity of 1 day. This is due to the daily update of the forecast wind data set: it introduces a spurious artificial periodicity that perfectly correlates with the daily $S1$-component tidal variability. The observed temporal decay (with correlation time scale of approximately 5 h) is well reproduced by the DET model $u_o$ auto-correlation from FP9, while it is overestimated by FP5. This can be easily explained: FP5 has a 12 h forcing time scale that brings spurious memory to the system.

The time decay and the modulation of the SUP $u_o$ and $v_o$ temporal auto-correlation functions are well reproduced in both simulations. The fact that the SUP initial decay is comparable with the HFR pattern shows that the value of the $\gamma_u$ coefficient shown in Table 4 is well chosen. The amplitudes in FP5 underestimate the observed ones, while in FP9 the daily peaks are present. In both the simulations the SUP auto-correlation is more damped in time with respect to the DET one, due to the introduction of the stochasticity, as expected. The long time variability of the correlation functions of the idealized models is





not expected to be similar to the observed ones, since the model space domain is zero-dimensional and the equations do not include the dynamics of eddies and other coherent two-dimensional structures.

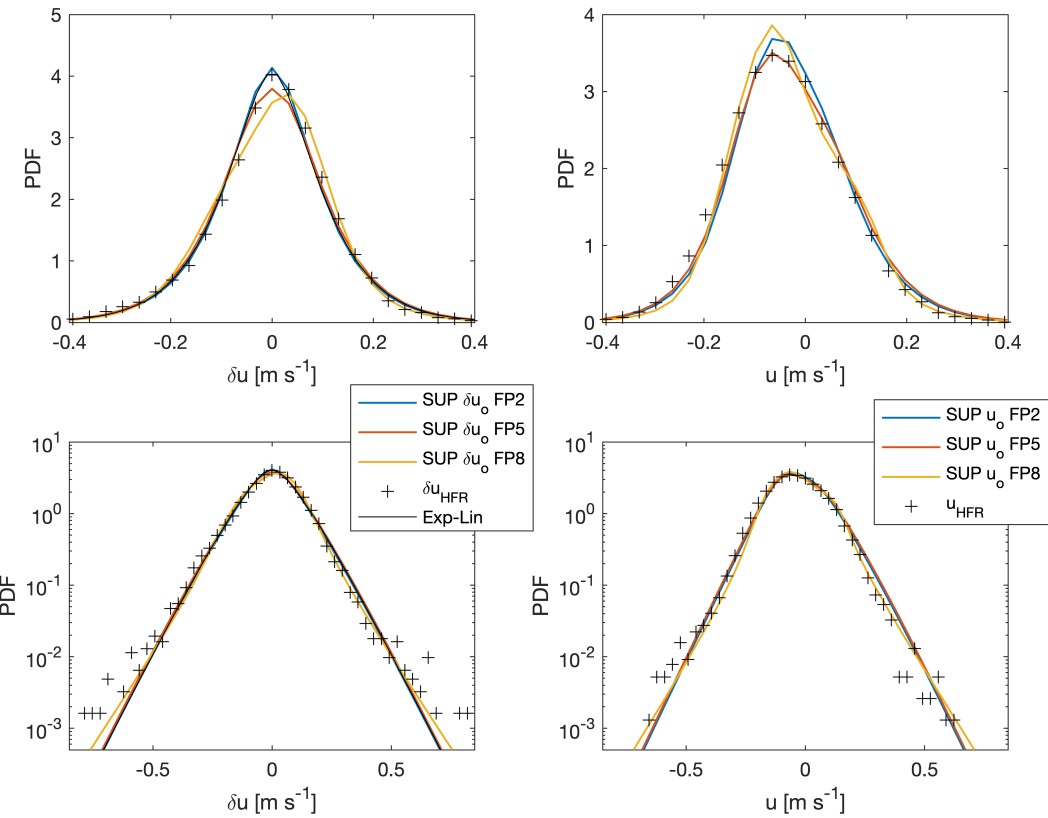

**Figure 8.** PDFs of the HFR (black crosses) and SUP model total velocity increments $\delta u$ (left column) and sea surface velocities $u$ (right column) in lin-lin plots (top row) and log-lin plots (bottom row) from: FP2 (blue line), FP5 (orange line) and FP8 (yellow line). The analytical superstatistical PDF from Eq. (6) is also reported (black line).

We have seen that moving from longer forcing time averaging FP5 to no forcing averaging FP9, the stochastic model needs less dof. In particular, the number of stochastic dof must decrease when the unaveraged tidal components are considered, as reported in Table 2. This can be seen numerically in Fig. 8, where all the PDFs are able to fit appropriately the observations. The rest of the FPs presented in Table 2 (FP3, FP4, FP6 and FP7) are not shown, because of the following reasons. FP4 and FP7 do not take into account any wind forcing, so they are not able to mimic the observed velocities. FP3 and FP6, looking at

the PDFs of SUP $\delta u_o$ and $u_o$, give very similar results to FP5, where the main differences are in the height of the PDFs peaks.



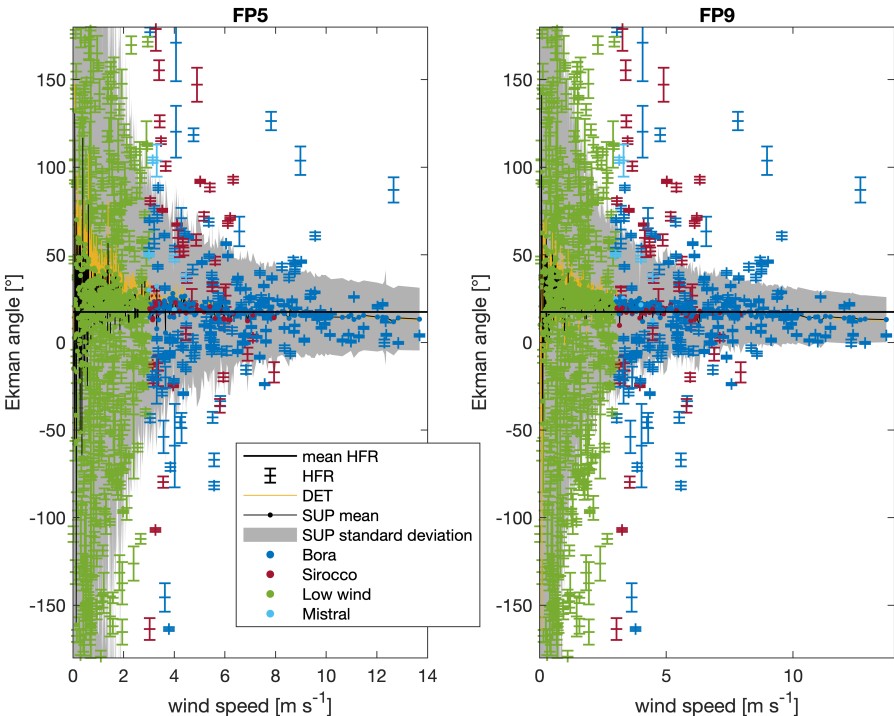

**Figure 9.** Ekman angle as a function of the daily WRF wind speed using the HFR (sticks with errorabars) and the models sea surface currents from FP5 (left) and FP9 (right): DET model Ekman angle (yellow line) and SUP model ensemble mean Ekman angle (black line with coloured dots, the colours indicate the present wind regime). The grey area indicates the SUP model ensemble standard deviation for the Ekman angle. The errors are calculated as shown in Appendix D. The horizontal black line is the observed HFR Ekman angle average.

In order to check if the models capture the observed veering angle, the angle between the daily averaged wind and the daily averaged selected sea surface current is computed (a positive Ekman angle means that the sea surface current is on the right with respect to the wind). As seen in Fig. 9, the observed HFR Ekman angle shows a large spread for low wind regimes, while it accumulates around the observed mean value with stronger wind speeds. In both simulations the Ekman angle obtained

from the DET model (yellow line) collapses towards the same angle range for high wind speeds but with less variability with respect to the observations. The SUP mean Ekman angle is almost the same as for the DET model and converges towards the observed mean Ekman angle with increasing wind speed. The SUP modelled standard deviation (grey area) reflects the observed variability on the entire domain of the Ekman angle for low wind speeds and shows a decrease of variability for stronger wind speeds. The SUP modelled Ekman angle includes, most of the times, the observed Ekman angles inside the one

standard deviation band. FP5 shows, both for the DET and SUP model, higher variability bands with respect to FP9 due to the higher number of stochastic dof.



## 4.2 The FRR and the SUP model

In this section the FRR is tested on the SUP model under the forcings of FP9. In Fig. 10 the perturbation methodology on the time series, described in Sect. 3.2, and the effects on the total sea surface velocity are visualized. In this figure, it is the $u_S$ velocity component to be perturbed, while the independent $v_S$ component remains unperturbed. The total SUP $u_o$ velocity component is affected by a clearly visible perturbation that decays in time, while the total SUP $v_o$ velocity component is slightly perturbed through the Ekman drag term.

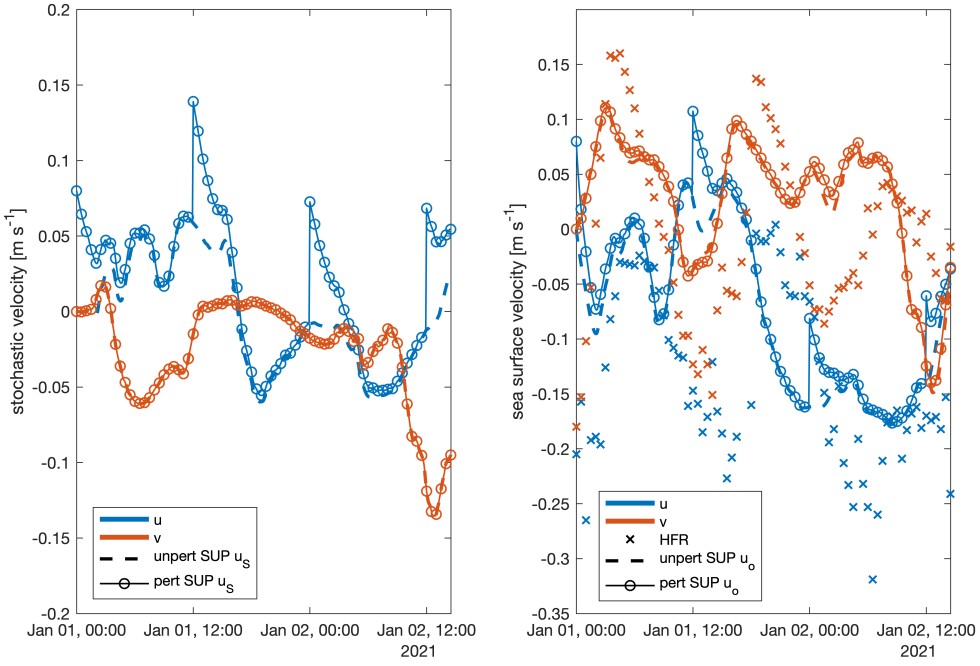

**Figure 10.** SUP model FP9 stochastic velocities $u_S$ (left); HFR (crosses) and SUP FP9 total sea surface velocities $u_o$ (right): $u$-component in blue, $v$-component in red. The SUP model time series are from the first walker in the: unperturbed system (dashed lines) and perturbed system (continuous lines with circles).

In Fig. 11 the diagonal response functions $R_{uu}(t)$ and $R_{vv}(t)$ with the correlations of the observed, unperturbed Ekman, stochastic and total velocities and analytical functions discussed further are reported. The diagonal response functions show an exponential decay with time scale corresponding to the stochastic velocity drag coefficient $\gamma_u$ (i.e. $1/\gamma_u \simeq 140$ min, see exponential fit in Fig. 11). When it is $u_E$ or $v_E$ that is perturbed, the diagonal response functions show an exponential decay with time scale corresponding to the mean Ekman drag term $\frac{C_B \tilde{u}}{\tilde{h}}$ (i.e. $\frac{\tilde{h}}{C_B \tilde{u}} \simeq 65$ min, not shown). This reveals that any external perturbation is decaying due to the corresponding model drag ($\gamma_u$ if we perturbe the stochastic velocity, $\frac{C_B \tilde{u}}{\tilde{h}}$ if we perturbe the Ekman velocity).




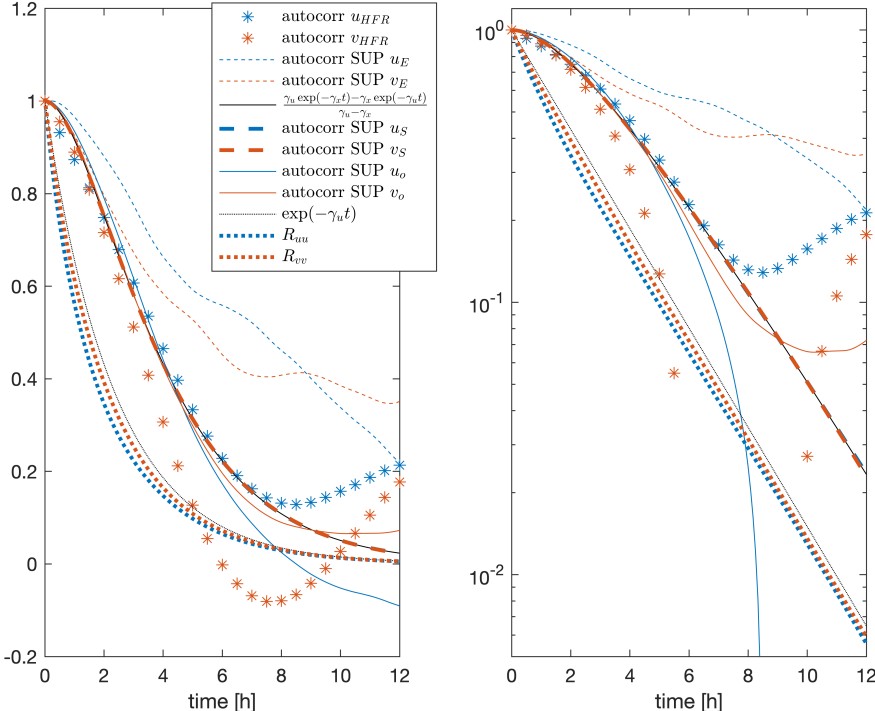

**Figure 11.** Diagonal response functions (thick dotted lines) and temporal autocorrelations in lin-lin plots (left) and lin-log plots (right) of HFR (asterisks) and unperturbed SUP model FP9 velocities: $u_E$ (dashed and dotted line), $u_S$ (thick dashed line), $u_o$ (continuous line). The blue indicates the $u$-component, while the red the $v$-component. The analytical autocorrelation function of the stochastic velocity from Wirth (2019) is reported with the black continuous line. The exponential decay with the stochastic drag coefficient $\gamma_u$ is shown with the black thin dotted line.

As expected, since $u_o$ and $v_o$ are not Gaussian variables and include terms that are not affected by the perturbation, as the tidal signal, the diagonal response functions' behaviour is different from the corresponding auto-correlations. It is then not possible to describe the response functions as linear combinations of the correlations in the unperturbed system.

What is particularly interesting to observe in Fig. 11 is that the SUP $u_S$ and $v_S$ auto-correlations coincide with the analytical auto-correlation function for a stochastic Gaussian variable, defined through a linear SDE with coloured Gaussian noise, found by Wirth (2019) in his Appendix C2:

$$C_{u_S}(t) = \frac{\gamma_u e^{-\gamma_x t} - \gamma_x e^{-\gamma_u t}}{\gamma_u - \gamma_x}, \tag{15}$$

showing a dependence on the drag coefficients only and not on the variance of the coloured Gaussian. This result seems surprising, since our stochastic velocities are obtained through a linear SDE with Bessel, and not Gaussian, coloured noise. Our interpretation of the fact is the following: the stochastic velocity drag time scale $1/\gamma_u \simeq 140$ min is one order of magnitude



smaller than the superstatistical gaussianity time scale $T \simeq 2$ days of the superstatistical analysis. This means that, under the

correlation point of view, the stochastic velocity does not have the time to develop the non-gaussian characteristics, but operates
at an almost constant variance. As a consequence, after the decay time of about 5 h, the SUP $u_S$ and $v_S$ auto-correlations
collapse exponentially with the same $\gamma_u$ coefficient of the diagonal response functions. This can be checked also analytically,
through Eq. (15) and knowing that $\gamma_u < \gamma_x$:

$$C_{u_S}(t) = \frac{\gamma_x}{\gamma_u - \gamma_x}\left(\frac{\gamma_u}{\gamma_x}e^{-\gamma_x t} - e^{-\gamma_u t}\right) \quad \xrightarrow{t \to \infty} \quad \frac{\gamma_x}{\gamma_x - \gamma_u}e^{-\gamma_u t} \quad \propto \quad R_{uu}(t) \tag{16}$$

So, knowing the auto-correlation function of the stochastic velocity, it is possible to obtain the response function of the total
velocity. It is then possible to state that the FRR holds in the SUP model when the perturbation is applied to the stochastic
signal.

## 4.3 Predictability evaluation results

In Fig. 12 the data assimilation perturbation methodology on the time series, described in Sect. 3.3, and the effects on the total
sea surface velocities are visualized. In this figure, it is the $u_S$ and $v_S$ velocity components to be perturbed. Both the total SUP
$u_o$ and $v_o$ show the data assimilation perturbation subject to an evident decay in time.

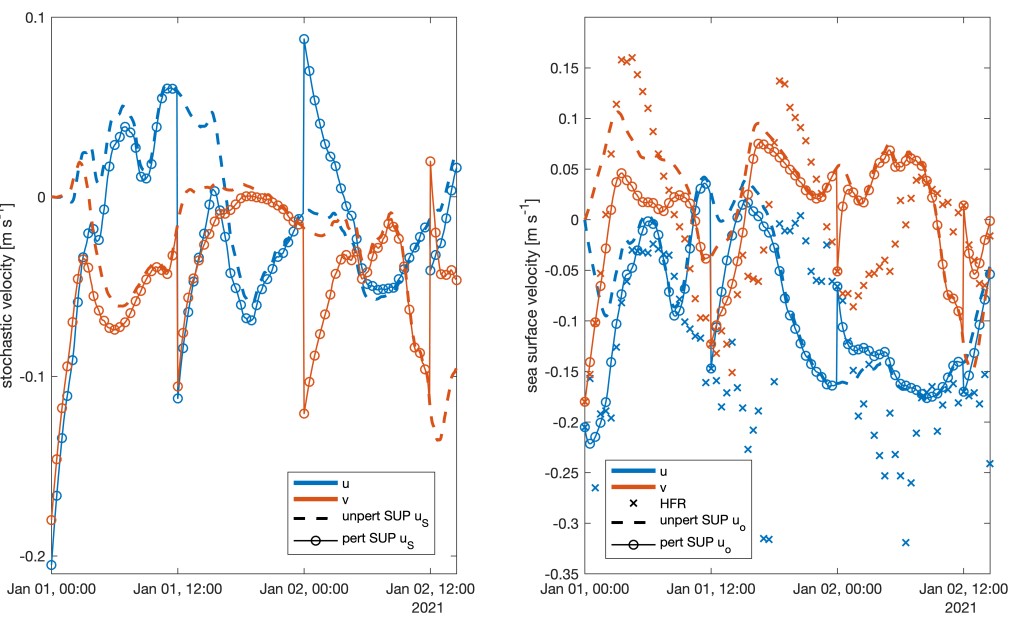

**Figure 12.** SUP model FP9 stochastic velocities (left); HFR (crosses) and SUP FP9 total sea surface velocities (right): $u$-component in blue,
$v$-component in red. The SUP model time series are from the first walker in the unperturbed system (dashed lines) and perturbed system
using a data-assimilation method (continuous lines with circles).





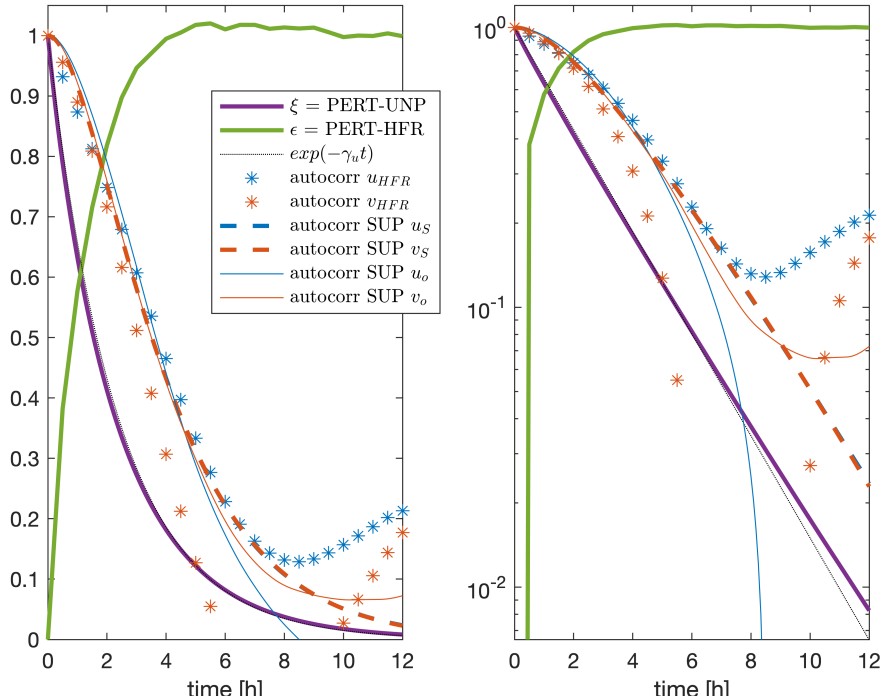

**Figure 13.** Temporal autocorrelations in lin-lin (left) and lin-log plots (right) of the HFR (asterisks) and SUP model stochastic (thick dashed lines) and total (continuous lines) $u$ (blue) and $v$ (red) sea surface velocities with the $\xi$ (purple thick continuous line) and $\epsilon$ (green thick continuous line) functions. The exponential fits of the $\xi$ function are shown (purple thin continuous lines). The exponential decay with the stochastic drag coefficient $\gamma_u$ is shown with the black thin dotted line.

The $\xi(t)$ and $\epsilon(t)$ functions are reported in Fig. 13 and represent the normailzed distance between the perturbed-unperturbed and the perturbed-observed systems, respectively. The function $\xi(t)$ can be considered as a generalization of the response function in the FRR when both the components are perturbed of a quantity not known a priori. It is possible to see that the $\xi(t)$ function has an exponential decay with time scale $1/\gamma_u \simeq 140$ min. When it is the Ekman signal to be perturbed, we obtain two exponential decays: $\tau_1 \simeq 75$ min for $t < 2$ h and $\tau_2 \simeq 140$ min for $t > 4$ h (not shown). The external perturbation, due to the observed data assimilation, decays with the stochastic drag coefficient when we perturbe the stochastic velocity, while a clear correspondance with the drag coefficient is not found when the Ekman signal is perturbed. The $\epsilon(t)$ function increases its value in time from zero and, after about 5 hours, saturates to one, showing that after that time the perturbed system has completely lost the memory of the initialization to the observations. This time of 5 hours coincides with the auto-correlation time scale of the SUP stochastic and total velocities.



## 5 Conclusions

In the Gulf of Trieste the analytical superstatistical PDF of the observed HFR sea surface current increments is known (Flora
et al., 2023) and is here called Exp-Lin PDF. The Exp-Lin PDF is non-Gaussian with fat-tails (the extreme events occur more
often with respect to a Gaussian statistics), it provides all the statistical moments but does not give any predictive information.
In this study we have developed a hierarcy of idealized models whose aim is to mimic the sea surface current time series,
characterized by the observed statistics.

The hierarchy is organized as follows: the DET model is purely deterministic and includes tidal and Ekman signals, the GAU
model adds a coloured-in-time Gaussian stochastic velocity, while the SUP model adds a stochastic sea surface velocity with
superstatistical fat-tailed increments that simulate the unresolved dynamics. This is done using Langevin equations in the
GAU model, whose solutions are Gaussian distributed Ornstein-Uhlenbeck processes, and modified Langevin equations with
coloured non-Gaussian noise in the SUP model, whose solutions are superstatistical variables. A variable is superstatistical if it
is locally (in time) Gaussian with a variance evolving over a longer time scale. In our case the variance is obtained through the
sum of $2\nu$ squared Ornstein-Uhlenbeck processes and results to be Gamma distributed with a shape parameter equal to $\nu$. The
parameter $\nu$ identifies the dof of the stochastic system. The models are then tested imposing nine FPs combining different types
of tidal and wind forcings: complete, 12 h moving-averaged time series or omitted forcing (Table 2). The stochatic models are
adjusted for each FP in order to best fit the observed PDF.

In the following we point out the general differences between the models' results. The DET model is able to simulate the
slow variability of the observed HFR currents, reaching the 55-56% of correlation coefficient (Fig. 4). The GAU and SUP
stochastic models increase the variability to observed values, but, since their ensemble means almost coincide with the DET
model results, they show little influence on the slow dynamics (Fig. 3).

The main differences between the models are evident looking at the PDFs of their variables. Regarding the total velocity incre-
ment PDFs (Fig. 5), the DET model does not explain the variability of the observed superstatistical Exp-Lin PDF, which results
in a stark underestimation of the variability. The GAU model improves the results, but without the observed fat tails, that is
the occurence of extreme events is underestimated. When we consider the SUP model, in order to fit the observed statistics,
the stochastic modelization requires $\nu = 2$ stochastic dof when the tidal forcing is omitted or averaged, while it needs $\nu = 1/2$
stochastic dof when the complete tidal signal is considered (Table 2). This result affects the shape of the SUP stochastic ve-
locity increment PDF (Fig. 2) that is Exp-Lin when $\nu = 2$ and Bessel when $\nu = 1/2$. The SUP model shows a total velocity
increment PDF that is representative of observations (Fig. 5): it is almost Exp-Lin when the FP considers averaged forcings and
$\nu = 2$ stochastic dof, while it has a shifted and slightly lower peak but tails that satisfactorily fit the observed data when the FP
consideres the complete forcings and $\nu = 1/2$ stochastic dof. The convergence to the PDF and, in particular, the enlargement
of the PDF tails to the observed ones in the progression from the DET model to the GAU and then to the SUP model is also
visible in the PDFs of the velocities (Fig. 6). In addition, we obtain that the FP with complete forcings (FP9) and, consequently,
with $\nu = 1/2$ stochastic dof, enables the SUP model to catch the tails with a double slope in the velocity PDF, as seen from the



observations.

The velocities temporal auto-correlation functions under the FP9 (Fig. 7) show for both the DET and the SUP models a realistic
time decay of around 5 h, but for longer times the stochasticity allows for a better representation in amplitude of successive
modulations, mitigating the underestimates and overestimates of the DET model peaks. In relation to Ekman dynamics (Fig. 9),
both the DET model and the SUP model ensemble mean replicate the tendency to form an angle between the wind and the sea
surface current that closely corresponds to the time averaged observed value. The SUP model shows a large standard deviation
under low wind forcing, which decreases with stronger wind forcing. This indicates that strong wind increases the likelihood of
the SUP model generating trajectories that match the observed average Ekman state. This pattern aligns with the HFR Ekman
angle data, where lower wind forcing results in more variability, while higher wind forcing brings values closer to the average.

The SUP model under the most complete forcings and, consequently, with $\nu = 1/2$ dof (FP9) is then explored applying exter-
nal perturbations to the stochastic signal. The response function trend can be obtained as the time limit (with time that tends
to infinity) of the stochastic velocity auto-correlation, see Fig. 11 and Eq. (16). The implication is that the FRR holds when
the stochastic signal is perturbed, showing that the response to an external perturbation can be obtained by considering the
fluctuations of the unperturbed system. In particular, the SUP model response function decays with the stochastic velocity drag
time scale ($1/\gamma_u$), indicating a clear memory time scale for external stochastic perturbations.
The simplified data assimilation exercise shows that the SUP model, given any external correction to the HFR values, reaches
an equilibrium distance from the observations, converges to the unperturbed realization, after 5 hours (Fig. 13). This time value
is consistent with the auto-correlation decay of the observed and SUP stochastic and total velocities.

From the predictive point of view, we should remark that our modelling is extremely idealised: it is local in space and does
not take into account other external forcings besides wind and tides, such as river discharge. The modelling can be further
developed including the Isonzo/Soča river influence, other air-sea interaction processes as wave generation, moving the space
domain from one point (0 dimension) to 2 dimensions including advection, eddy dynamics and also incorporating the broader
Adriatic circulation.
Despite its idealization, the SUP model presented in this study can reproduce part of the deterministic (tide-wind-forced) large
scale dynamics and the observed fat tails of the HFR velocity increments PDFs and is eventually able to simulate extreme
events. One of the most interesting results is that when we have a more detailed deterministic signal (given by the complete
forcings), a larger part of the dynamics is resolved and the stochastic part of the model must not only have less variability, it
must also decrease its dof. The SUP model can therefore be taken as an example for the modelling community to reflect on
how stochasticity can be used to reproduce extreme events and how its characteristics vary depending on the coarse graining
of the forcings. Various applications can be found to the SUP model, for example it is currently used to explore the role of
extreme events in the kinetic energy fluxes between the atmosphere and the sea.




*Code and data availability.* The exact version of the model used to produce the results used in this paper is archived on Zenodo (Flora et al., 2024), as are input data and scripts to run the model for all the simulations presented in this paper. In particular: `twd_model.f90` is the code implementing the DET model, `twg_model.f90` is the code implementing the GAU model, `tws_model.f90` is the code implementing the STO model with $\nu = 2$ and `twb_model.f90` is the code implementing the STO model with $\nu = 1/2$. Additionally, `twb_FRRD_model.f90`, `twb_FRRS_model.f90`, `twb_datassD_model.f90` and `twb_datassS_model.f90` are the codes implementing the STO model with $\nu = 1/2$ in the FRR and data assimilation applications (applied to the deterministic or stochastic signals). The HFR sea surface current data of the Gulf of Trieste are freely available from the European HFRadar node (OGS et al., 2023). The WRF forecasted wind field is obtainable upon request from ARPA FVG (https://www.arpa.fvg.it, last access: 29 October 2024, CRMA).


## Appendix A: An analytical idealized case

This is the idealized case of the stochastic system described in Sect. 3, Eq. (2) and Eq. (3) in which:

$$\sqrt{\Sigma_{i=1}^{2\nu}\alpha_i^2} = \sqrt{Q} \simeq \text{constant} \tag{A1}$$

This approximation and the following analysis permits to find the analytical distribution of the variables $x$, $u_S$ and $\delta u_S$. Considering the $u$-components, independent from the $v$-components, Eq. (2) and Eq. (3) can be written as follows:

$$
\begin{cases}
dx = -\gamma_x x \, dt + \sqrt{Q} \, dW_x \\
du_S = -\gamma_u u_S \, dt + \eta x \, dt
\end{cases}
$$

$$
d\begin{pmatrix} x \\ u_S \end{pmatrix} = \begin{pmatrix} -\gamma_x & 0 \\ \eta & -\gamma_u \end{pmatrix} \begin{pmatrix} x \\ u_S \end{pmatrix} dt + \begin{pmatrix} \sqrt{Q} \\ 0 \end{pmatrix} dW_x \tag{A2}
$$

$$
= M \begin{pmatrix} x \\ u_S \end{pmatrix} dt + \begin{pmatrix} \sqrt{Q} \\ 0 \end{pmatrix} dW_x
$$



where $M = \begin{pmatrix} -\gamma_x & 0 \\ \eta & -\gamma_u \end{pmatrix}$ can be diagonalized:

$$det(M - mI) = det \begin{pmatrix} -\gamma_x - m & 0 \\ \eta & -\gamma_u - m \end{pmatrix} = (\gamma_x + m)(\gamma_u + m) = 0 \tag{A3}$$

obtaining two eigenvalues $m_1 = -\gamma_x$ and $m_2 = -\gamma_u$ and the following eigenvectors (with $\gamma_x \neq \gamma_u$):

$Me_1 = m_1 e_1$

$$\begin{pmatrix} -\gamma_x & 0 \\ \eta & -\gamma_u \end{pmatrix} \begin{pmatrix} e_{1x} \\ e_{1y} \end{pmatrix} = -\gamma_x \begin{pmatrix} e_{1x} \\ e_{1y} \end{pmatrix}$$

$Me_2 = m_2 e_2$

$$\begin{cases} -\gamma_x e_{1x} = -\gamma_x e_{1x} \\ \eta e_{1x} - \gamma_u e_{1y} = -\gamma_x e_{1y} \end{cases}$$

$$\begin{pmatrix} -\gamma_x & 0 \\ \eta & -\gamma_u \end{pmatrix} \begin{pmatrix} e_{2x} \\ e_{2y} \end{pmatrix} = -\gamma_u \begin{pmatrix} e_{2x} \\ e_{2y} \end{pmatrix}$$

$$\begin{cases} e_{1y} = -\frac{\eta}{\gamma_x - \gamma_u} e_{1x} \\ e_{1x}^2 + e_{1y}^2 = 1 \end{cases}$$

$$\begin{cases} -\gamma_x e_{2x} = -\gamma_u e_{2x} \\ \eta e_{2x} - \gamma_u e_{2y} = -\gamma_u e_{2y} \end{cases} \tag{A4}$$

$$e_{1x}^2 \left(1 + \frac{\eta^2}{(\gamma_x - \gamma_u)^2}\right) = 1$$

$$\begin{cases} e_{2x} = 0 \\ e_{2y} = 1 \end{cases}$$

$$\begin{cases} e_{1x} = -\frac{\gamma_x - \gamma_u}{\sqrt{\eta^2 + (\gamma_x - \gamma_u)^2}} \\ e_{1y} = \frac{\eta}{\sqrt{\eta^2 + (\gamma_x - \gamma_u)^2}} \end{cases}$$

$$e_2 = \begin{pmatrix} 0 \\ 1 \end{pmatrix}$$

$$e_1 = \begin{pmatrix} -\frac{\gamma_x - \gamma_u}{\sqrt{\eta^2 + (\gamma_x - \gamma_u)^2}} \\ \frac{\eta}{\sqrt{\eta^2 + (\gamma_x - \gamma_u)^2}} \end{pmatrix}$$

For $\gamma_x \neq \gamma_u$, the system in Eq. (A2) is diagonalizable and it is possible to write $M = T \cdot D \cdot T^{-1}$ where:

$$T = \begin{pmatrix} -\frac{\gamma_x - \gamma_u}{\sqrt{\eta^2 + (\gamma_x - \gamma_u)^2}} & 0 \\ \frac{\eta}{\sqrt{\eta^2 + (\gamma_x - \gamma_u)^2}} & 1 \end{pmatrix} \qquad D = \begin{pmatrix} -\gamma_x & 0 \\ 0 & -\gamma_u \end{pmatrix} \qquad T^{-1} = \begin{pmatrix} -\frac{\sqrt{\eta^2 + (\gamma_x - \gamma_u)^2}}{\gamma_x - \gamma_u} & 0 \\ \frac{\eta}{\gamma_x - \gamma_u} & 1 \end{pmatrix} \tag{A5}$$

and so, hypothizing $\gamma_u < \gamma_x$ (the final distributions do not change with $\gamma_x < \gamma_u$, what is changing is just the sign of the Wiener
processes in the diagonalised system):

$$d \begin{pmatrix} x \\ u_S \end{pmatrix} = T \cdot D \cdot T^{-1} \begin{pmatrix} x \\ u_S \end{pmatrix} dt + \begin{pmatrix} \sqrt{Q} \\ 0 \end{pmatrix} dW_x \tag{A6}$$

$$d \left[ T^{-1} \begin{pmatrix} x \\ u_S \end{pmatrix} \right] = D \left[ T^{-1} \begin{pmatrix} x \\ u_S \end{pmatrix} \right] dt + T^{-1} \begin{pmatrix} \sqrt{Q} \\ 0 \end{pmatrix} dW_x \tag{A7}$$



$$
\quad \begin{pmatrix} \tilde{x} \\ \tilde{u_S} \end{pmatrix} = T^{-1} \begin{pmatrix} x \\ u_S \end{pmatrix} = \begin{pmatrix} -\frac{\sqrt{\eta^2+(\gamma_x-\gamma_u)^2}}{\gamma_x-\gamma_u} x \\ u_S + \frac{\eta}{\gamma_x-\gamma_u} x \end{pmatrix} ; \qquad \begin{pmatrix} -\sqrt{Q_x} \\ \sqrt{Q_u} \end{pmatrix} = T^{-1} \begin{pmatrix} \sqrt{Q} \\ 0 \end{pmatrix} = \begin{pmatrix} -\frac{\sqrt{\eta^2+(\gamma_x-\gamma_u)^2}}{\gamma_x-\gamma_u} \sqrt{Q} \\ \frac{\eta}{\gamma_x-\gamma_u} \sqrt{Q} \end{pmatrix} \tag{A8}
$$

obtaining:

$$
d \begin{pmatrix} \tilde{x} \\ \tilde{u_S} \end{pmatrix} = D \begin{pmatrix} \tilde{x} \\ \tilde{u_S} \end{pmatrix} dt + \begin{pmatrix} -\sqrt{Q_x} \\ \sqrt{Q_u} \end{pmatrix} dW_x
$$

$$
\begin{cases} d\tilde{x} = -\gamma_x \tilde{x} dt - \sqrt{Q_x} dW_x \\ d\tilde{u_S} = -\gamma_u \tilde{u_S} dt + \sqrt{Q_u} dW_x \end{cases} \tag{A9}
$$

The approximation for which $\sqrt{Q} = $ constant leads to $\sqrt{Q_x}$ and $\sqrt{Q_u}$ also constants. This fact leads to the Gaussianity of the variables $\tilde{x}$ and $\tilde{u_S}$ (they are Ornstein-Uhlenbeck processes):

$$
\quad \begin{cases} \tilde{x} \sim N\left(0, \sigma_{\tilde{x}}^2\right) = N\left(0, \frac{Q_x}{2\gamma_x}\right) = N\left(0, \frac{\eta^2+(\gamma_x-\gamma_u)^2}{(\gamma_x-\gamma_u)^2} \frac{Q}{2\gamma_x}\right) \\ \tilde{u_S} \sim N\left(0, \sigma_{\tilde{u_S}}^2\right) = N\left(0, \frac{Q_u}{2\gamma_u}\right) = N\left(0, \frac{\eta^2}{(\gamma_x-\gamma_u)^2} \frac{Q}{2\gamma_u}\right) \end{cases} \tag{A10}
$$

From Eq. (A8) it is possible to write $x$ and $u_S$ in terms of $\tilde{x}$ and $\tilde{u_S}$, resulting as linear combinations of Gaussian variables:

$$
\begin{cases} x = -\frac{\gamma_x-\gamma_u}{\sqrt{\eta^2+(\gamma_x-\gamma_u)^2}} \tilde{x} \\ u_S = \tilde{u_S} + \tilde{r} \\ \tilde{r} = -\frac{\eta}{\gamma_x-\gamma_u} x = \frac{\eta}{\sqrt{\eta^2+(\gamma_x-\gamma_u)^2}} \tilde{x} \end{cases} \tag{A11}
$$

It leads to the Gaussianity of the variables $x$, $\tilde{r}$ and $u_S$ with the following variances:

$$
\begin{cases} x \sim N\left(0, \sigma_x^2\right) \\ \tilde{r} \sim N\left(0, \sigma_{\tilde{r}}^2\right) \\ u_S \sim N\left(0, \sigma_{u_S}^2\right) \end{cases} \tag{A12}
$$


$$
\begin{cases} \sigma_x^2 &= \frac{(\gamma_x-\gamma_u)^2}{\eta^2+(\gamma_x-\gamma_u)^2} \sigma_{\tilde{x}}^2 = \frac{Q}{2\gamma_x} \\ \sigma_{\tilde{r}}^2 &= \frac{\eta^2}{\eta^2+(\gamma_x-\gamma_u)^2} \sigma_{\tilde{x}}^2 = \frac{\eta^2}{(\gamma_x-\gamma_u)^2} \frac{Q}{2\gamma_x} \\ \sigma_{u_S}^2 &= \sigma_{\tilde{u_S}}^2 + \sigma_{\tilde{r}}^2 + 2 \lim_{t\to\infty} \mathrm{corr}(\tilde{u_S}(t), \tilde{r}(t)) \\ &= \frac{\eta^2}{(\gamma_x-\gamma_u)^2} \frac{Q}{2\gamma_u} + \frac{\eta^2}{(\gamma_x-\gamma_u)^2} \frac{Q}{2\gamma_x} + 2 \frac{\eta}{\sqrt{\eta^2+(\gamma_x-\gamma_u)^2}} \lim_{t\to\infty} \mathrm{corr}(\tilde{u_S}(t), \tilde{x}(t)) \\ &= \frac{\eta^2(\gamma_x+\gamma_u)}{(\gamma_x-\gamma_u)^2} \frac{Q}{2\gamma_x\gamma_u} - \frac{2\eta^2}{(\gamma_x-\gamma_u)^2(\gamma_x+\gamma_u)} Q \\ &= \frac{\eta^2}{(\gamma_x-\gamma_u)^2} \frac{(\gamma_x+\gamma_u)^2-4\gamma_x\gamma_u}{2\gamma_x\gamma_u(\gamma_x+\gamma_u)} Q \\ &= \frac{\eta^2}{2\gamma_x\gamma_u(\gamma_x+\gamma_u)} Q \end{cases} \tag{A13}
$$





where, since $\tilde{x}$ and $\tilde{u_S}$ are Ornstein-Uhlenbeck processes:

$$\begin{cases} \tilde{x}(t) = -e^{-\gamma_x t} \int_0^t e^{\gamma_x t'} \sqrt{Q_x} F(t') dt' \\ \tilde{u_S}(t) = e^{-\gamma_u t} \int_0^t e^{\gamma_u t'} \sqrt{Q_u} F(t') dt' \\ E[F(t')F(t'')] = \delta(t' - t'') \end{cases} \tag{A14}$$

we have that:

$$\begin{aligned} \text{corr}(\tilde{u_S}(t), \tilde{x}(t)) &= -e^{-(\gamma_x + \gamma_u)t} \sqrt{Q_x Q_u} \int_0^t \int_0^t e^{\gamma_x t'} e^{\gamma_u t''} E[F(t')F(t'')] dt' dt'' \\ &= -\frac{\eta \sqrt{\eta^2 + (\gamma_x - \gamma_u)^2}}{(\gamma_x - \gamma_u)^2} Q e^{-(\gamma_x + \gamma_u)t} \frac{e^{(\gamma_x + \gamma_u)t} - 1}{\gamma_x + \gamma_u} \\ &= -\frac{\eta \sqrt{\eta^2 + (\gamma_x - \gamma_u)^2}}{(\gamma_x - \gamma_u)^2} \frac{Q}{\gamma_x + \gamma_u} \left(1 - e^{-(\gamma_x + \gamma_u)t}\right) \\ &\xrightarrow{t \to \infty} -\frac{\eta \sqrt{\eta^2 + (\gamma_x - \gamma_u)^2}}{(\gamma_x - \gamma_u)^2} \frac{Q}{\gamma_x + \gamma_u} \end{aligned} \tag{A15}$$

Now, it is possible to express the $\delta u_S$ variable, starting from its definition and using Eq. (A13), in terms of the diagonalized variables $\tilde{x}$ and $\tilde{u_S}$:

$$\begin{aligned} \delta u_S(t) &= u_S(t + \delta) - u_S(t) \\ &= \tilde{u_S}(t + \delta) + \frac{\eta}{\sqrt{\eta^2 + (\gamma_x - \gamma_u)^2}} \tilde{x}(t + \delta) - \tilde{u_S}(t) - \frac{\eta}{\sqrt{\eta^2 + (\gamma_x - \gamma_u)^2}} \tilde{x}(t) \\ &= [\tilde{u_S}(t + \delta) - \tilde{u_S}(t)] + \frac{\eta}{\sqrt{\eta^2 + (\gamma_x - \gamma_u)^2}} [\tilde{x}(t + \delta) - \tilde{x}(t)] \\ &= \delta\tilde{u_S}(t) + \frac{\eta}{\sqrt{\eta^2 + (\gamma_x - \gamma_u)^2}} \delta\tilde{x}(t) \end{aligned} \tag{A16}$$

where:

$$\begin{cases} \delta\tilde{u_S}(t) = \tilde{u_S}(t + \delta) - \tilde{u_S}(t) \\ \delta\tilde{x}(t) = \tilde{x}(t + \delta) - \tilde{x}(t) \end{cases} \tag{A17}$$





Since $\delta\tilde{u}_S$ and $\delta\tilde{x}$ are linear combinations of Gaussian variables, they are Gaussian variables themselves:

$$
\begin{cases}
\delta\tilde{u}_S \sim N(0, \sigma^2_{\delta\tilde{u}_S}) \\
\sigma^2_{\delta\tilde{u}_S} = 2\sigma^2_{\tilde{u}_S} + 2\lim_{t\to\infty}\mathrm{corr}\left(\tilde{u}_S(t+\delta), \tilde{u}_S(t)\right) \\
\qquad = 2\dfrac{\eta^2}{(\gamma_x - \gamma_u)^2}\dfrac{Q}{2\gamma_u}\left(1 - e^{-\gamma_u\delta}\right) \\
\delta\tilde{x} \sim N(0, \sigma^2_{\delta\tilde{x}}) \\
\sigma^2_{\delta\tilde{x}} = 2\sigma^2_{\tilde{x}} + 2\lim_{t\to\infty}\mathrm{corr}\left(\tilde{x}(t+\delta), \tilde{x}(t)\right) \\
\qquad = 2\dfrac{\eta^2 + (\gamma_x - \gamma_u)^2}{(\gamma_x - \gamma_u)^2}\dfrac{Q}{2\gamma_x}\left(1 - e^{-\gamma_x\delta}\right)
\end{cases}
\tag{A18}
$$

where:

$$
\begin{aligned}
\mathrm{corr}\left(\tilde{u}_S(t+\delta), \tilde{u}_S(t)\right) &= \frac{\eta^2 Q}{(\gamma_x - \gamma_u)^2}\, e^{-2\gamma_u t}\, e^{-\gamma_u\delta}\int_0^{t+\delta}\int_0^{t} e^{\gamma_u t'}e^{\gamma_u t''}\delta(t' - t'')\,dt'\,dt'' \\
&= \frac{\eta^2 Q}{(\gamma_x - \gamma_u)^2}\, e^{-2\gamma_u t}\, e^{-\gamma_u\delta}\int_0^{t} e^{2\gamma_u t'}\,dt' \\
&= \frac{\eta^2 Q}{(\gamma_x - \gamma_u)^2}\, e^{-\gamma_u\delta}\, e^{-2\gamma_u t}\frac{e^{2\gamma_u t} - 1}{2\gamma_u} \\
&= \frac{\eta^2 Q}{(\gamma_x - \gamma_u)^2}\frac{e^{-\gamma_u\delta}}{2\gamma_u}\left(1 - e^{-2\gamma_u t}\right) \\
&\xrightarrow{t\to\infty} \frac{\eta^2}{(\gamma_x - \gamma_u)^2}\frac{Q}{2\gamma_u}\, e^{-\gamma_u\delta}
\end{aligned}
\tag{A19}
$$


$$
\begin{aligned}
\mathrm{corr}\left(\tilde{x}(t+\delta), \tilde{x}(t)\right) &= \frac{\eta^2 + (\gamma_x - \gamma_u)^2}{(\gamma_x - \gamma_u)^2}\, Q\, e^{-2\gamma_x t}\, e^{-\gamma_x\delta}\int_0^{t+\delta}\int_0^{t} e^{\gamma_x t'}e^{\gamma_x t''}\delta(t' - t'')\,dt'\,dt'' \\
&= \frac{\eta^2 + (\gamma_x - \gamma_u)^2}{(\gamma_x - \gamma_u)^2}\, Q\, e^{-2\gamma_x t}\, e^{-\gamma_x\delta}\int_0^{t} e^{2\gamma_x t'}\,dt' \\
&= \frac{\eta^+ (\gamma_x - \gamma_u)^2}{(\gamma_x - \gamma_u)^2}\, Q\, e^{-\gamma_x\delta}\, e^{-2\gamma_x t}\frac{e^{2\gamma_x t} - 1}{2\gamma_x} \\
&= \frac{\eta^2 + (\gamma_x - \gamma_u)^2}{(\gamma_x - \gamma_u)^2}\frac{Q}{2\gamma_x}\, e^{-\gamma_x\delta}\left(1 - e^{-2\gamma_x t}\right) \\
&\xrightarrow{t\to\infty} \frac{\eta^2 + (\gamma_x - \gamma_u)^2}{(\gamma_x - \gamma_u)^2}\frac{Q}{2\gamma_x}\, e^{-\gamma_x\delta}
\end{aligned}
\tag{A20}
$$



Now, according to Eq. (A16) and Eq. (A18), since $\delta u_S$ is a linear combination of Gaussian variables, it is Gaussianly distributed itself:

$$\begin{cases} \delta u_S \sim N(0, \sigma_{\delta u}^2) \\ \sigma_{\delta u}^2 = \sigma_{\tilde{\delta u}}^2 + \frac{\eta^2}{\eta^2 + (\gamma_x - \gamma_u)^2}\, \sigma_{\tilde{\delta x}}^2 + \frac{2\eta}{\sqrt{\eta^2 + (\gamma_x - \gamma_u)^2}} \lim_{t \to \infty} \mathrm{corr}(\tilde{\delta u_S}(t), \tilde{\delta x}(t)) \end{cases} \tag{A21}$$

where:

$$\lim_{t \to \infty} \mathrm{corr}(\tilde{\delta u_S}(t), \tilde{\delta x}(t)) = \lim_{t \to \infty} \mathrm{corr}\left(\tilde{u_S}(t+\delta) - \tilde{u_S}(t), \tilde{x}(t+\delta) - \tilde{x}(t)\right)$$

$$= \lim_{t \to \infty} \left[\mathrm{corr}\left(\tilde{u_S}(t+\delta), \tilde{x}(t+\delta)\right) - \mathrm{corr}\left(\tilde{u_S}(t+\delta), \tilde{x}(t)\right) - \mathrm{corr}\left(\tilde{u_S}(t), \tilde{x}(t+\delta)\right) + \mathrm{corr}\left(\tilde{u_S}(t), \tilde{x}(t)\right)\right] \tag{A22}$$

$$\lim_{t \to \infty} \mathrm{corr}\left(\tilde{u_S}(t+\delta), \tilde{x}(t+\delta)\right) = \lim_{t \to \infty} \mathrm{corr}\left(\tilde{u_S}(t), \tilde{x}(t)\right)$$

$$= -\frac{\eta \sqrt{\eta^2 + (\gamma_x - \gamma_u)^2}}{(\gamma_x - \gamma_u)^2} \frac{Q}{\gamma_x + \gamma_u} \tag{A23}$$

$$\mathrm{corr}\left(\tilde{u_S}(t+\delta), \tilde{x}(t)\right) = -\frac{\eta \sqrt{\eta^2 + (\gamma_x - \gamma_u)^2}}{(\gamma_x - \gamma_u)^2}\, Q\, e^{-\gamma_u(t+\delta)}\, e^{-\gamma_x t} \int_0^{t+\delta} \int_0^t e^{\gamma_u t'} e^{\gamma_x t''} \delta(t' - t'') dt' dt''$$

$$= -\frac{\eta \sqrt{\eta^2 + (\gamma_x - \gamma_u)^2}}{(\gamma_x - \gamma_u)^2}\, Q\, e^{-\gamma_u \delta}\, e^{-(\gamma_x + \gamma_u)t} \int_0^t e^{(\gamma_x + \gamma_u)t'} dt'$$

$$= -\frac{\eta \sqrt{\eta^2 + (\gamma_x - \gamma_u)^2}}{(\gamma_x - \gamma_u)^2}\, \frac{Q}{\gamma_x + \gamma_u}\, e^{-\gamma_u \delta} \left(1 - e^{-(\gamma_x + \gamma_u)t}\right)$$

$$\xrightarrow{t \to \infty} -\frac{\eta \sqrt{\eta^2 + (\gamma_x - \gamma_u)^2}}{(\gamma_x - \gamma_u)^2}\, \frac{Q}{\gamma_x + \gamma_u}\, e^{-\gamma_u \delta} \tag{A24}$$


$$\mathrm{corr}\left(\tilde{u_S}(t), \tilde{x}(t+\delta)\right) = -\frac{\eta \sqrt{\eta^2 + (\gamma_x - \gamma_u)^2}}{(\gamma_x - \gamma_u)^2}\, Q\, e^{-\gamma_u t}\, e^{-\gamma_x(t+\delta)} \int_0^{t+\delta} \int_0^t e^{\gamma_u t'} e^{\gamma_x t''} \delta(t' - t'') dt' dt''$$

$$\xrightarrow{t \to \infty} -\frac{\eta \sqrt{\eta^2 + (\gamma_x - \gamma_u)^2}}{(\gamma_x - \gamma_u)^2}\, \frac{Q}{\gamma_x + \gamma_u}\, e^{-\gamma_x \delta} \tag{A25}$$

And so, continuing Eq. (A22):

$$\lim_{t \to \infty} \mathrm{corr}\left(\tilde{u_S}(t+\delta), \tilde{x}(t+\delta)\right) = -\frac{\eta \sqrt{\eta^2 + (\gamma_x - \gamma_u)^2}}{(\gamma_x - \gamma_u)^2}\, \frac{Q}{\gamma_x + \gamma_u} \left[2 - \left(e^{-\gamma_x \delta} + e^{-\gamma_u \delta}\right)\right] \tag{A26}$$



It is now possible the calculation started in Eq. (A21):

$$
\begin{aligned}
\sigma_{\delta u}^2 =& \sigma_{\tilde{\delta}u}^2 + \frac{\eta^2}{\eta^2 + (\gamma_x - \gamma_u)^2}\, \sigma_{\tilde{\delta}x}^2 + \frac{2\eta}{\sqrt{\eta^2 + (\gamma_x - \gamma_u)^2}}\, \lim_{t\to\infty} \mathrm{corr}(\delta\tilde{u}_S(t), \tilde{\delta}x(t)) \\
=& \frac{2\eta^2}{(\gamma_x - \gamma_u)^2}\, \frac{Q}{2\gamma_u}\left(1 - e^{-\gamma_u \delta}\right) + \frac{\eta^2}{\eta^2 + (\gamma_x - \gamma_u)^2}\, \frac{2\left[\eta^2 + (\gamma_x - \gamma_u)^2\right]}{(\gamma_x - \gamma_u)^2}\, \frac{Q}{2\gamma_x}\left(1 - e^{-\gamma_x \delta}\right) + \\
& + \frac{2\eta}{\sqrt{\eta^2 + (\gamma_x - \gamma_u)^2}}\,(-)\,\frac{\eta\sqrt{\eta^2 + (\gamma_x - \gamma_u)^2}}{(\gamma_x - \gamma_u)^2}\, \frac{Q}{\gamma_x + \gamma_u}\left[2 - \left(e^{-\gamma_x \delta} + e^{-\gamma_u \delta}\right)\right] \\
=& \frac{2\eta^2 Q}{(\gamma_x - \gamma_u)^2}\left[\frac{1}{2\gamma_u}\left(1 - e^{-\gamma_u \delta}\right) + \frac{1}{2\gamma_x}\left(1 - e^{-\gamma_x \delta}\right) - \frac{1}{\gamma_x + \gamma_u}\left(1 - e^{-\gamma_x \delta}\right) - \frac{1}{\gamma_x + \gamma_u}\left(1 - e^{-\gamma_u \delta}\right)\right] \\
=& \frac{2\eta^2 Q}{(\gamma_x - \gamma_u)^2}\left[\frac{\gamma_x - \gamma_u}{2\gamma_u(\gamma_x + \gamma_u)}\left(1 - e^{-\gamma_u \delta}\right) - \frac{\gamma_x - \gamma_u}{2\gamma_x(\gamma_x + \gamma_u)}\left(1 - e^{-\gamma_x \delta}\right)\right] \\
=& \frac{\eta^2 Q}{\gamma_x^2 - \gamma_u^2}\left(\frac{1 - e^{-\gamma_u \delta}}{\gamma_u} - \frac{1 - e^{-\gamma_x \delta}}{\gamma_x}\right)
\end{aligned}
\tag{A27}
$$





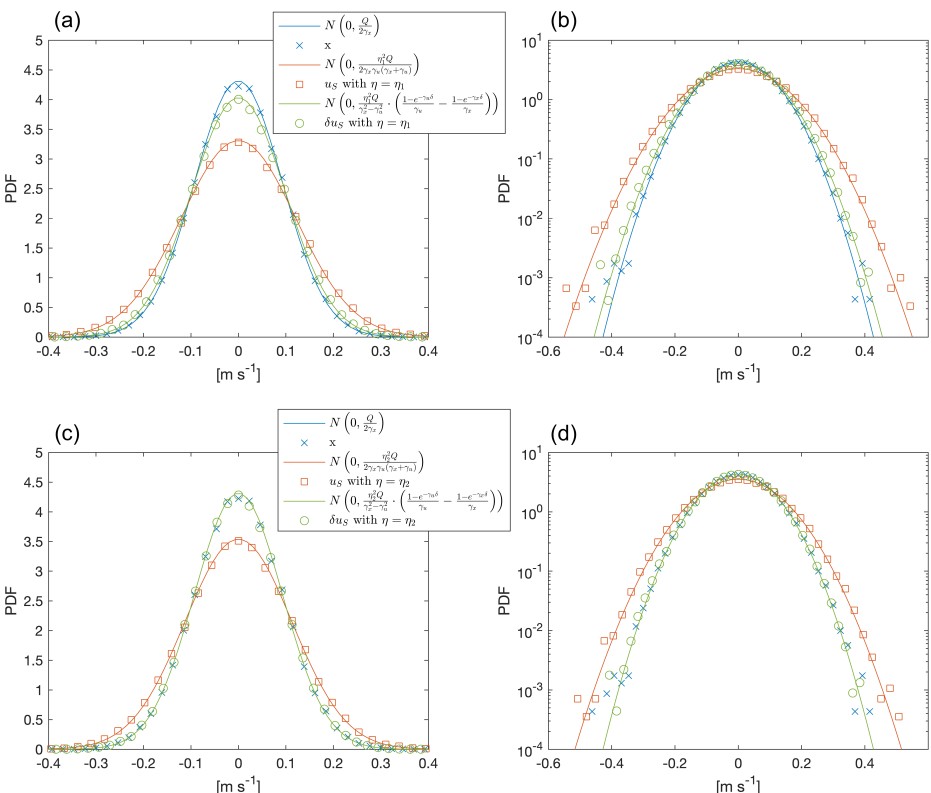

**Figure A1.** Gaussian PDFs of the stochastic variables $x$ (in blue), $u_S$ (in red) and $\delta u_S$ (in green): the continuous lines are the analytical Gaussian PDFs in Eq. (A28, Eq. A29 and Eq. A30), while the points are the histograms of the simulated variables described in Eq. (A2) and Eq. (A16) in the case $Q =$ constant. The used parameters are shown in Table A1, the simulation with $\eta = \eta_1$ results are shown in (a) and (b), while in (c) and (d) the results with $\eta = \eta_2$ are shown. In (a) and (c) the plots are lin-lin, while in (b) and (d) the plots are lin-log.





| parameter | value |
|---|---|
| ensemble size | $10^5$ |
| total integration time | $\sim 656$ days |
| time step $dt$ | $1.5 \times 10^2$ s |
| $\tau$ | $6.6 \times 10^3$ s |
| $\gamma_x = 1/\tau$ | $1.515 \times 10^{-4}$ s$^{-1}$ |
| $Q$ | $2.6 \times 10^{-6}$ m$^2$s$^{-3}$ |
| $\gamma_u$ | $5.152 \times 10^{-5}$ s$^{-1}$ |
| $\eta_1$ | $1.333 \times 10^{-4}$ s$^{-1}$ |
| $\eta_2 = \sqrt{\frac{\gamma_x^2 - \gamma_u^2}{2\left[\frac{\gamma_x}{\gamma_u}(1-e^{-\gamma_u\delta})-(1-e^{-\gamma_x\delta})\right]}}$ | $1.247 \times 10^{-4}$ s$^{-1}$ |
| $\delta$ | $1.44 \times 10^4$ s |

**Table A1.** Involved parameters in the idealized analytical stochastic modelization.

In a summary, in the idealized case of the stochastic system, in which $Q$ is constant, the variables $x$, $u_S$ and $\delta u_S$ are Gaussianly distributed:

$$x \sim N\left(0, \sigma_x^2\right) = N\left(0, \frac{Q}{2\gamma_x}\right) \tag{A28}$$


$$u_S \sim N\left(0, \sigma_{u_S}^2\right) = N\left(0, \frac{\eta^2 Q}{2\gamma_x \gamma_u(\gamma_x + \gamma_u)}\right) \tag{A29}$$

$$\delta u_S \sim N\left(0, \sigma_{\delta u}^2\right) = N\left(0, \frac{\eta^2 Q}{\gamma_x^2 - \gamma_u^2}\left(\frac{1-e^{-\gamma_u\delta}}{\gamma_u} - \frac{1-e^{-\gamma_x\delta}}{\gamma_x}\right)\right) \tag{A30}$$

It is possible to find a condition for which $\delta u_S$ is distributed as $x$:

$$
\begin{aligned}
\sigma_{\delta u}^2 &= \frac{\eta^2 Q}{\gamma_x^2 - \gamma_u^2}\left(\frac{1-e^{-\gamma_u\delta}}{\gamma_u} - \frac{1-e^{-\gamma_x\delta}}{\gamma_x}\right) \\
&= \frac{Q}{2\gamma_x}\frac{2\eta^2}{\gamma_x^2 - \gamma_u^2}\left[\frac{\gamma_x}{\gamma_u}\left(1-e^{-\gamma_u\delta}\right) - \left(1-e^{-\gamma_x\delta}\right)\right] \\
&= \sigma_x^2 \frac{2\eta^2}{\gamma_x^2 - \gamma_u^2}\left[\frac{\gamma_x}{\gamma_u}\left(1-e^{-\gamma_u\delta}\right) - \left(1-e^{-\gamma_x\delta}\right)\right]
\end{aligned}
\tag{A31}
$$


$$
\begin{aligned}
\sigma_{\delta u}^2 = \sigma_x^2 &\iff \frac{2\eta^2}{\gamma_x^2 - \gamma_u^2}\left[\frac{\gamma_x}{\gamma_u}\left(1-e^{-\gamma_u\delta}\right) - \left(1-e^{-\gamma_x\delta}\right)\right] = 1 \\
&\iff \eta = \sqrt{\frac{\gamma_x^2 - \gamma_u^2}{2\left[\frac{\gamma_x}{\gamma_u}\left(1-e^{-\gamma_u\delta}\right) - \left(1-e^{-\gamma_x\delta}\right)\right]}}
\end{aligned}
\tag{A32}
$$





All these facts are confirmed by the numerical simulations results, shown in Fig. A1 using the numerical parameters in Table A1.

**Appendix B: Ornstein–Uhlenbeck processes and colored noise**

Let's consider the stochastic part for the $u$-component (analogously for the $v$-component) of the SUP model:

$$d\alpha_i = -\mu\alpha_i dt + \beta_u dW_i \tag{B1}$$

$$dx = -\gamma_x x dt + \sqrt{\Sigma_{i=1}^{2\nu}\alpha_i^2}\, dW_x \tag{B2}$$


$$du_S = -\gamma_u u_S dt + \eta x dt \tag{B3}$$

Since $\beta$ and $\mu$ are constants, Eq. (B1) is a Langevin equation and the variables $\alpha_i$ are Ornstein–Uhlenbeck processes, so their distribution in the stationary state is Gaussian:

$$p_\alpha(\alpha_i) = \sqrt{\frac{\mu}{\pi\beta_u^2}} e^{-\mu\alpha_i^2/\beta_u^2} \tag{B4}$$

Defining the new variable $Q$ as the sum of $2\nu$ squared identically distributed Gaussian variables $Q = \Sigma_{i=1}^{2\nu}\alpha_i^2$, it results as a variable distributed with a Gamma distribution with shape parameter equal to $\nu$ in the stationary state:

$$\begin{aligned}
p_Q(Q) &= \Gamma_{\nu,\tilde{\lambda}}(Q) \\
&= \frac{\tilde{\lambda}^\nu Q^{\nu-1} e^{-\tilde{\lambda}Q}}{\Gamma(\nu)} \\
&= \left(\frac{\mu}{\beta_u^2}\right)^\nu \frac{Q^{\nu-1}}{\Gamma(\nu)} e^{-\mu Q/\beta_u^2}
\end{aligned} \tag{B5}$$

where $\tilde{\lambda} = \frac{\mu}{\beta_u^2}$ and with first moment:

$$\langle Q \rangle = \int_0^\infty \left(\frac{\mu}{\beta^2}\right)^\nu \frac{Q^\nu}{\Gamma(\nu)} e^{-\mu Q/\beta^2} dQ = \frac{\nu\beta^2}{\mu}. \tag{B6}$$

It is possible to write Eq. (B2) in the following way:

$$dx = -\gamma_x x dt + \sqrt{Q}dW_x \tag{B7}$$

If $Q$ were a constant, then the equation above would be a Langevin equation and $x$ would represent an Ornstein-Uhlenbeck process. But for construction $Q$ is slowly varying respect $x$ (because $\tau = \frac{1}{\gamma_x} \ll T = \frac{1}{\mu}$). So it is possible to assume that locally





in time $Q$ is constant and so locally $x$ has a Gaussian distribution, given a certain variance $\sigma_x^2 = \frac{Q}{2\gamma_x}$ determined by the value

of $Q$:

$$
\begin{aligned}
p(x|\sigma_x^2) &= N\left(0, \sigma_x^2\right) \\
&= \frac{1}{\sqrt{2\pi\sigma_x^2}} e^{-x^2/2\sigma_x^2} \\
&= \sqrt{\frac{\gamma_x}{\pi Q}}\, e^{-\gamma_x x^2/Q}
\end{aligned}
\tag{B8}
$$

In order to obtain the total $x$ PDF, the $\sigma_x^2$ PDF $f(\sigma_x^2)$ is needed. Thanks to its dependence with $Q$, it is possible to obtain it:

$$
f(\sigma_x^2) = \Gamma_{\nu,\lambda}(\sigma_x^2) = \frac{\lambda^\nu \sigma_x^{2(\nu-1)} e^{-\lambda\sigma_x^2}}{\Gamma(\nu)} = \left(\frac{2\mu\gamma_x}{\beta_u^2}\right)^\nu \frac{\sigma_x^{2(\nu-1)}}{\Gamma(\nu)}\, e^{-\frac{2\mu\gamma_x}{\beta_u^2}\sigma_x^2}
\tag{B9}
$$

where $\lambda = \frac{2\mu\gamma_x}{\beta_u^2}$. It follows that:

$$
\begin{aligned}
p_\nu(x) &= \int_0^\infty f(\sigma_x^2) p(x|\sigma_x^2)\, d\sigma_x^2 \\
&= \frac{\lambda^\nu}{\Gamma(\nu)\sqrt{2\pi}} \int_0^\infty \sigma_x^{2(\nu-1)} e^{-\lambda\sigma_x^2} \frac{1}{\sqrt{\sigma_x^2}} e^{-x^2/2\sigma_x^2}\, d(\sigma_x^2) \\
&= \frac{\lambda^\nu}{\Gamma(\nu)\sqrt{2\pi}} \int_0^\infty z^{\nu-3/2} e^{-\lambda z - x^2/2z}\, dz
\end{aligned}
\tag{B10}
$$

that, for $\nu = 2$, it gives the following result (Gradshteyn and Ryzhik (2014) page 369, Eq. (3.471.16), called, in this article, Exp-Lin PDF:

$$
\begin{aligned}
p_{\nu=2}(x) &= -\frac{\lambda^2}{\sqrt{2}} \frac{\partial}{\partial\lambda}\left(\lambda^{-1/2} e^{-\sqrt{2\lambda}|x|}\right) \\
&= \frac{\sqrt{2\lambda}\, e^{-\sqrt{2\lambda}|x|}(\sqrt{2\lambda}|x| + 1)}{4} \\
&= \frac{\sqrt{\mu\gamma_x}}{2\beta_u} e^{-2\sqrt{\mu\gamma_x}|x|/\beta_u}\left(\frac{2\sqrt{\mu\gamma_x}}{\beta_u}|x| + 1\right)
\end{aligned}
\tag{B11}
$$

while, for $\nu = 1/2$ (Gradshteyn and Ryzhik (2014) page 370, Eq. 3.478.4):

$$
\begin{aligned}
p_{\nu=\frac{1}{2}}(x) &= \frac{\sqrt{2\lambda}}{\pi} K_0\left(\sqrt{2\lambda}|x|\right) \\
&= \frac{2\sqrt{\mu\gamma_x}}{\pi\beta_u} K_0\left(\frac{2\sqrt{\mu\gamma_x}}{\beta_u}|x|\right)
\end{aligned}
\tag{B12}
$$

where $K_0(z)$ is the modified Bessel function of the second kind of 0 order. A summary of the results of this Appendix for the cases $\nu$, 2 and 1/2 degrees of freedom (dof) is given in Table B1. Finally, Eq. (B3) does not present any analytic solution.



| dof | $p_Q(Q)$ | $\langle Q \rangle$ | $p(x)$ |
|---|---|---|---|
| $\nu$ | $\left(\frac{\mu}{\beta_u^2}\right)^\nu \frac{Q^{\nu-1}}{\Gamma(\nu)} e^{-\mu Q/\beta_u^2}$ | $\nu\beta^2/\mu$ | $\frac{(2\mu\gamma_x/\beta^2)^\nu}{\Gamma(\nu)\sqrt{2\pi}} \int_0^\infty z^{\nu-3/2} e^{-\frac{2\mu\gamma_x}{\beta^2}z - \frac{x^2}{2z}}\, dz$ |
| 2 | $\left(\frac{\mu}{\beta_u^2}\right)^2 Q\, e^{-\mu Q/\beta_u^2}$ | $2\beta^2/\mu$ | $\frac{\sqrt{\mu\gamma_x}}{2\beta_u} e^{-2\sqrt{\mu\gamma_x}|x|/\beta_u} \left(\frac{2\sqrt{\mu\gamma_x}}{\beta_u}|x|+1\right)$ |
| $\frac{1}{2}$ | $\sqrt{\frac{\mu}{\pi\beta_u^2}} \frac{e^{-\mu Q/\beta_u^2}}{\sqrt{Q}}$ | $\beta^2/2\mu$ | $\frac{2\sqrt{\mu\gamma_x}}{\pi\beta_u} K_0\left(\frac{2\sqrt{\mu\gamma_x}}{\beta_u}|x|\right)$ |

**Table B1.** Analytical PDFs and means of the variable $Q$ and analytical PDFs of the variable $x$ for $\nu$, 2 and 1/2 degrees of freedom.

**Appendix C: The Ekman system of the deterministic model - the coefficients linear relation with the wind speed**

Defining the energy $E = \tilde{h}\frac{|\boldsymbol{u_o}|^2}{2}$ for the DET model in Eq. (1) and supposing, as a first approximation, the tidal signal negligible
$\boldsymbol{u_o} \simeq \boldsymbol{u_E}$ (daily time scale), a slowlying varying sea surface layer depth $\partial_t\tilde{h} = 0$ and neglecting the eddy depletion term
$\boldsymbol{F} = \rho_a c_a |\boldsymbol{u_a}|\boldsymbol{u_a}$, so the energy variation in time can be expressed as follows:

$$\partial_t E = \tilde{h}\,\partial_t \frac{|\boldsymbol{u_E}|^2}{2} = \tilde{h}\,(u_E\partial_t u_E + v_E\partial_t v_E) = -C_B|\boldsymbol{u_E}|^3 + u_E F_u + v_E F_v \tag{C1}$$

from which:

$$\frac{C_B}{\tilde{h}} = \frac{u_E F_u + v_E F_v}{\tilde{h}|\boldsymbol{u_E}|^3} - \frac{\partial_t|\boldsymbol{u_E}|^2}{2|\boldsymbol{u_E}|^3} \tag{C2}$$

Discretizing Eq. (C1) and Eq. (C2) around the time step $t_i$ (where $\Delta t = t_{i+1} - t_i$) and using the centered scheme for the
temporal derivative $\partial_t|\boldsymbol{u_E}|^2 = \frac{|\boldsymbol{u_E}(t_{i+1})|^2 - |\boldsymbol{u_E}(t_{i-1})|^2}{2\Delta t}$), it results:

$$\tilde{h}\,\frac{|\boldsymbol{u_E}(t_{i+1})|^2 - |\boldsymbol{u_E}(t_{i-1})|^2}{4} = -C_B(t_i)\,|\boldsymbol{u_E}(t_i)|^3\Delta t + [u_E(t_i)\,F_u(t_i) + v_E(t_i)\,F_v(t_i)]\,\Delta t \tag{C3}$$

and:

$$\begin{aligned}\frac{C_B(t_i)}{\tilde{h}} &= \frac{u_E(t_i)\,F_u(t_i) + v_E(t_i)\,F_v(t_i)}{\tilde{h}|\boldsymbol{u_E}(t_i)|^3} - \frac{|\boldsymbol{u_E}(t_{i+1})|^2 - |\boldsymbol{u_E}(t_{i-1})|^2}{4|\boldsymbol{u_E}(t_i)|^3\Delta t} \\ &= \frac{\alpha(t_i)}{\tilde{h}} - \zeta(t_i)\end{aligned} \tag{C4}$$

where $\alpha(t_i) = \frac{u_E(t_i)\,F_u(t_i) + v_E(t_i)\,F_v(t_i)}{|\boldsymbol{u_E}(t_i)|^3}$, $\zeta(t_i) = \frac{|\boldsymbol{u_E}(t_i+\Delta t)|^2 - |\boldsymbol{u_E}(t_i-\Delta t)|^2}{4|\boldsymbol{u_E}(t_i)|^3\Delta t}$ and $\tilde{h}$ has a low dependence on the time. Approx-
imating the daily-averaged observed sea surface current as the daily deterministic Ekman sea surface current, it is possible to
calculate the coefficients $\alpha(t_i)$ and $\zeta(t_i)$ directly from the observations and the daily-averaged wind forcing.





Eq. (1) results in:

$$\begin{pmatrix} \partial_t u_E \\ \partial_t v_E \end{pmatrix} = \begin{pmatrix} -\left(\frac{\alpha}{\tilde{h}}-\zeta\right)|\boldsymbol{u_E}| & f \\ -f & -\left(\frac{\alpha}{\tilde{h}}-\zeta\right)|\boldsymbol{u_E}| \end{pmatrix} \begin{pmatrix} u_E \\ v_E \end{pmatrix} + \begin{pmatrix} \frac{F_u}{\tilde{h}} \\ \frac{F_v}{\tilde{h}} \end{pmatrix} \tag{C5}$$


$$\begin{cases} \partial_t u_E = -\frac{\alpha}{\tilde{h}}|\boldsymbol{u_E}|u_E + \zeta|\boldsymbol{u_E}|u_E + fv_D + \frac{F_u}{\tilde{h}} \\ \partial_t v_E = -\frac{\alpha}{\tilde{h}}|\boldsymbol{u_E}|v_E + \zeta|\boldsymbol{u_E}|v_E - fu_E + \frac{F_v}{\tilde{h}} \end{cases} \tag{C6}$$

$$\begin{cases} \partial_t u_E - \zeta|\boldsymbol{u_E}|u_E - fv_E = \frac{F_u - \alpha|\boldsymbol{u_E}|u_E}{\tilde{h}} \\ \partial_t v_E - \zeta|\boldsymbol{u_E}|v_E + fu_E = \frac{F_v - \alpha|\boldsymbol{u_E}|v_E}{\tilde{h}} \end{cases} \tag{C7}$$

From which it is possible to obtain the slowly varying $\tilde{h}$:


$$\begin{cases} \tilde{h} = \frac{F_u - \alpha|\boldsymbol{u_E}|u_E}{\partial_t u_E - \zeta|\boldsymbol{u_E}|u_E - fv_E} \\ \tilde{h} = \frac{F_v - \alpha|\boldsymbol{u_E}|v_E}{\partial_t v_E - \zeta|\boldsymbol{u_E}|v_E + fu_E} \end{cases} \tag{C8}$$

The discretization gives:

$$\begin{cases} \tilde{h}(t_i) = \frac{F_u(t_i) - \alpha(t_i)\,|\boldsymbol{u_E}(t_i)|\,u_E(t_i)}{[u_E(t_i+\Delta t) - u_E(t_i-\Delta t)]/2\Delta t - \zeta(t_i)\,|\boldsymbol{u_E}(t_i)|\,u_E(t_i) - fv_D(t_i)} \\ \tilde{h}(t_i) = \frac{F_v(t_i) - \alpha(t_i)\,|\boldsymbol{u_E}(t_i)|\,v_E(t_i)}{[v_E(t_i+\Delta t) - v_E(t_i-\Delta t)]/2\Delta t - \zeta(t_i)\,|\boldsymbol{u_E}(t_i)|\,v_E(t_i) + fu_E(t_i)} \end{cases} \tag{C9}$$

The two expressions in Eq. (C9) should give a unique value for $\tilde{h}$ in function of the time directly from the daily-averaged observed currents and the daily-averaged wind forcing. In practice, they almost agree and we calculated $\tilde{h}(t_i)$ as the average

of these two expressions. From $\tilde{h}(t_i)$ it is possible to obtain also $C_B(t_i)$:

$$C_B(t_i) = \alpha(t_i) - \zeta(t_i) \cdot \tilde{h}(t_i) \tag{C10}$$

Having the $\tilde{h}$, the $C_B$ and the wind speed time series, it is possible to calculate the mean values of $\tilde{h}$ and $C_B$ in function of the wind speed, obtaining a linear fit after deleting the values exceeding one standard deviation (Fig. C1).



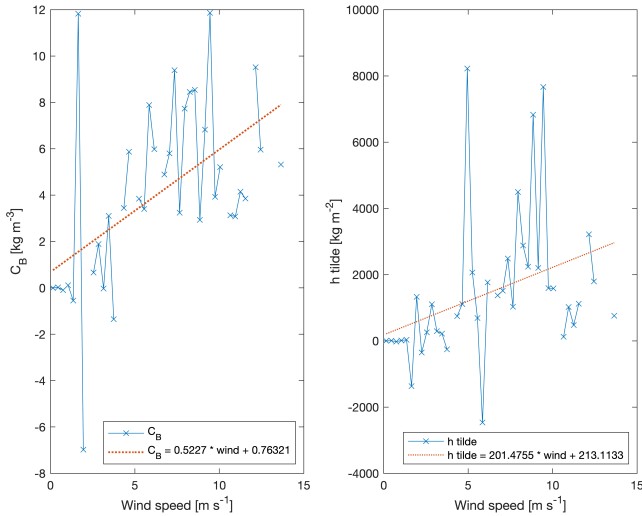

**Figure C1.** Dependence on the wind speed of the model coefficients $C_B$ and $\tilde{h}$, calculated accordingly to Eq. (C10) and to the mean of Eq.s (C9) respectively, and their linear regressions.

### Appendix D: On the Ekman angle standard deviation

In this study the Ekman angle is defined as the angle between the daily wind and the daily sea surface current, it is positive if the sea surface current is on the right respect the wind direction. For semplicity it is here assumed that the wind does not contribute with any error and is perfectly known. It follows that the standard deviation of the Ekman angle coincides with the standard deviation of the daily sea surface current angle. Under this assumption, the given Ekman angle standard deviation is an underestimation of its real error. So, in order to obtain this estimation of the Ekman angle standard deviation, starting

from the modelled (HFR measured) sea surface current components with 5 min (30 min) time resolution and their stochastic variability (measure accuracy), two points must be followed: (i) the daily average transformation and (ii) the polar coordinates transformation.

   Let's take into consideration the sea surface currents time series of a single day $x = \{u_1, ..., u_n, v_1, ..., v_n\}$, where $n = 288$

for the 5 min time resolution modelled sea surface currents and $n = 48$ for the 30 min HFR observed sea surface currents, with





their $2n \times 2n$ symmetric covariance matrix:

$$\Sigma^x = \begin{pmatrix} \sigma_{u_1}^2 & \cdots & \sigma_{u_1 u_n} & \sigma_{u_1 v_1} & \cdots & \sigma_{u_1 v_n} \\ \vdots & \ddots & \vdots & \vdots & & \vdots \\ \sigma_{u_n u_1} & \cdots & \sigma_{u_n}^2 & \sigma_{u_n v_1} & \cdots & \sigma_{u_n v_n} \\ \sigma_{v_1 u_1} & \cdots & \sigma_{v_1 u_n} & \sigma_{v_1}^2 & \cdots & \sigma_{v_1 v_n} \\ \vdots & & \vdots & \vdots & \ddots & \vdots \\ \sigma_{v_n u_1} & \cdots & \sigma_{v_n u_n} & \sigma_{v_n v_1} & \cdots & \sigma_{v_n}^2 \end{pmatrix} = \begin{pmatrix} \Sigma_{uu}^x & \Sigma_{uv}^x \\ \Sigma_{uv}^x & \Sigma_{vv}^x \end{pmatrix} \tag{D1}$$

Here the sea surface currents are the ensemble averaged values for each time and the covariance matrix values are the relative ensemble variances and covariances, while for the observations the sea surface currents are simply the indipendent HFR measured sea surface currents (each measurement is independent from the others) with their HFR accuracy and 0 covariances (so $\Sigma^x$ reduces to a diagonal matrix).

Calling $f = \begin{pmatrix} \bar{u} \\ \bar{v} \end{pmatrix}$ the daily average sea surface currents for a single day, its linear transformation (here the transformation matrix is called $A$) can be written as:

$$f = A \cdot x \tag{D2}$$


$$\begin{pmatrix} \bar{u} \\ \bar{v} \end{pmatrix} = \frac{1}{n} \begin{pmatrix} \overbrace{1\cdots1}^{n} & \overbrace{0\cdots0}^{n} \\ \underbrace{0\cdots0}_{n} & \underbrace{1\cdots1}_{n} \end{pmatrix} \begin{pmatrix} u_1 \\ \vdots \\ u_n \\ v_1 \\ \vdots \\ v_n \end{pmatrix} \tag{D3}$$





So, the covariance matrix of the transformed variable $\Sigma^f = \begin{pmatrix} \sigma_{\bar{u}}^2 & \sigma_{\bar{u}\bar{v}} \\ \sigma_{\bar{u}\bar{v}} & \sigma_{\bar{v}}^2 \end{pmatrix}$ can be obtained in the following way:

$$\Sigma^f = A \cdot \Sigma^x \cdot A^T$$

$$= \frac{1}{n^2} \begin{pmatrix} 1 & \cdots & 1 & 0 & \cdots & 0 \\ 0 & \cdots & 0 & 1 & \cdots & 1 \end{pmatrix} \begin{pmatrix} \sigma_{u_1}^2 & \cdots & \sigma_{u_1 u_n} & \sigma_{u_1 v_1} & \cdots & \sigma_{u_1 v_n} \\ \vdots & \ddots & \vdots & \vdots & & \vdots \\ \sigma_{u_n u_1} & \cdots & \sigma_{u_n}^2 & \sigma_{u_n v_1} & \cdots & \sigma_{u_n v_n} \\ \sigma_{v_1 u_1} & \cdots & \sigma_{v_1 u_n} & \sigma_{v_1}^2 & \cdots & \sigma_{v_1 v_n} \\ \vdots & & \vdots & \vdots & \ddots & \vdots \\ \sigma_{v_n u_1} & \cdots & \sigma_{v_n u_n} & \sigma_{v_n v_1} & \cdots & \sigma_{v_n}^2 \end{pmatrix} \begin{pmatrix} 1 & 0 \\ \vdots & \vdots \\ 1 & 0 \\ 0 & 1 \\ \vdots & \vdots \\ 0 & 1 \end{pmatrix}$$

$$= \frac{1}{n^2} \begin{pmatrix} 1 & \cdots & 1 & 0 & \cdots & 0 \\ 0 & \cdots & 0 & 1 & \cdots & 1 \end{pmatrix} \begin{pmatrix} \sigma_{u_1}^2 + \cdots + \sigma_{u_1 u_n} & \sigma_{u_1 v_1} + \cdots + \sigma_{u_1 v_n} \\ \vdots & \vdots \\ \sigma_{u_n u_1} + \cdots + \sigma_{u_n}^2 & \sigma_{u_n v_1} + \cdots + \sigma_{u_n v_n} \\ \sigma_{v_1 u_1} + \cdots + \sigma_{v_1 u_n} & \sigma_{v_1}^2 + \cdots + \sigma_{v_1 v_n} \\ \vdots & \vdots \\ \sigma_{v_n u_1} + \cdots + \sigma_{v_n u_n} & \sigma_{v_n v_1} + \cdots + \sigma_{v_n}^2 \end{pmatrix}$$

$$= \frac{1}{n^2} \begin{pmatrix} \sigma_{u_1}^2 + \cdots + \sigma_{u_1 u_n} + \cdots + \sigma_{u_n u_1} + \cdots + \sigma_n^2 & \sigma_{u_1 v_1} + \cdots + \sigma_{u_1 v_n} + \cdots + \sigma_{u_n v_1} + \cdots + \sigma_{u_n v_n} \\ \sigma_{v_1 u_1} + \cdots + \sigma_{v_1 u_n} + \cdots + \sigma_{v_n u_1} + \cdots + \sigma_{v_n u_n} & \sigma_{v_1}^2 + \cdots + \sigma_{v_1 v_n} + \cdots + \sigma_{v_n v_1} + \cdots + \sigma_{v_n}^2 \end{pmatrix}$$

$$= \frac{1}{n^2} \begin{pmatrix} \text{sum}(\Sigma_{uu}^x) & \text{sum}(\Sigma_{uv}^x) \\ \text{sum}(\Sigma_{uv}^x) & \text{sum}(\Sigma_{vv}^x) \end{pmatrix} \tag{D4}$$


Once obtained the daily sea surface current components $\begin{pmatrix} \bar{u} \\ \bar{v} \end{pmatrix}$ and their covariance matrix $\begin{pmatrix} \sigma_{\bar{u}}^2 & \sigma_{\bar{u}\bar{v}} \\ \sigma_{\bar{u}\bar{v}} & \sigma_{\bar{v}}^2 \end{pmatrix}$, it is possible to change from cartesian to polar coordinates:

$$\begin{cases} r = \sqrt{\bar{u}^2 + \bar{v}^2} \\ \theta = \arctan\left(\frac{\bar{v}}{\bar{u}}\right) \end{cases} \tag{D5}$$

The transformation now is not linear and one must resort to the jacobian matrix:

$$J = \begin{pmatrix} \frac{\partial r}{\partial \bar{u}} & \frac{\partial r}{\partial \bar{v}} \\ \frac{\partial \theta}{\partial \bar{u}} & \frac{\partial \theta}{\partial \bar{v}} \end{pmatrix} = \begin{pmatrix} \frac{\bar{u}}{\sqrt{\bar{u}^2 + \bar{v}^2}} & \frac{\bar{v}}{\sqrt{\bar{u}^2 + \bar{v}^2}} \\ -\frac{\bar{v}}{\bar{u}^2 + \bar{v}^2} & \frac{\bar{u}}{\bar{u}^2 + \bar{v}^2} \end{pmatrix} \tag{D6}$$




to obtain the new covariance matrix ($\theta$ and $\sigma_\theta$ are in radiants):

$$
\begin{pmatrix} \sigma_r^2 & \sigma_{r\theta} \\ \sigma_{r\theta} & \sigma_\theta^2 \end{pmatrix} = J \begin{pmatrix} \sigma_{\bar{u}}^2 & \sigma_{\bar{u}\bar{v}} \\ \sigma_{\bar{u}\bar{v}} & \sigma_{\bar{v}}^2 \end{pmatrix} J^T
$$

$$
= \begin{pmatrix} \frac{\bar{u}}{\sqrt{\bar{u}^2+\bar{v}^2}} & \frac{\bar{v}}{\sqrt{\bar{u}^2+\bar{v}^2}} \\ -\frac{\bar{v}}{\bar{u}^2+\bar{v}^2} & \frac{\bar{u}}{\bar{u}^2+\bar{v}^2} \end{pmatrix} \begin{pmatrix} \sigma_{\bar{u}}^2 & \sigma_{\bar{u}\bar{v}} \\ \sigma_{\bar{u}\bar{v}} & \sigma_{\bar{v}}^2 \end{pmatrix} \begin{pmatrix} \frac{\bar{u}}{\sqrt{\bar{u}^2+\bar{v}^2}} & -\frac{\bar{v}}{\bar{u}^2+\bar{v}^2} \\ \frac{\bar{v}}{\sqrt{\bar{u}^2+\bar{v}^2}} & \frac{\bar{u}}{\bar{u}^2+\bar{v}^2} \end{pmatrix}
$$

$$
= \begin{pmatrix} \frac{\bar{u}}{\sqrt{\bar{u}^2+\bar{v}^2}} & \frac{\bar{v}}{\sqrt{\bar{u}^2+\bar{v}^2}} \\ -\frac{\bar{v}}{\bar{u}^2+\bar{v}^2} & \frac{\bar{u}}{\bar{u}^2+\bar{v}^2} \end{pmatrix} \begin{pmatrix} \frac{\bar{u}\sigma_{\bar{u}}^2+\bar{v}\sigma_{\bar{u}\bar{v}}}{\sqrt{\bar{u}^2+\bar{v}^2}} & \frac{-\bar{v}\sigma_{\bar{u}}^2+\bar{u}\sigma_{\bar{u}\bar{v}}}{\bar{u}^2+\bar{v}^2} \\ \frac{\bar{u}\sigma_{\bar{u}\bar{v}}+\bar{v}\sigma_{\bar{v}}^2}{\sqrt{\bar{u}^2+\bar{v}^2}} & \frac{-\bar{v}\sigma_{\bar{u}\bar{v}}+\bar{u}\sigma_{\bar{v}}^2}{\bar{u}^2+\bar{v}^2} \end{pmatrix}
$$

$$
= \begin{pmatrix} \frac{\bar{u}^2\sigma_{\bar{u}}^2+2\bar{u}\bar{v}\sigma_{\bar{u}\bar{v}}+\bar{v}^2\sigma_{\bar{v}}^2}{\bar{u}^2+\bar{v}^2} & \frac{-\bar{u}\bar{v}\sigma_{\bar{u}}^2+\left(\bar{u}^2-\bar{v}^2\right)^2\sigma_{\bar{u}\bar{v}}+\bar{u}\bar{v}\sigma_{\bar{v}}^2}{(\bar{u}^2+\bar{v}^2)^{3/2}} \\ \frac{-\bar{u}\bar{v}\sigma_{\bar{u}}^2+\left(\bar{u}^2-\bar{v}^2\right)^2\sigma_{\bar{u}\bar{v}}+\bar{u}\bar{v}\sigma_{\bar{v}}^2}{(\bar{u}^2+\bar{v}^2)^{3/2}} & \frac{\bar{v}^2\sigma_{\bar{u}}^2-2\bar{u}\bar{v}\sigma_{\bar{u}\bar{v}}+\bar{u}^2\sigma_{\bar{v}}^2}{(\bar{u}^2+\bar{v}^2)^2} \end{pmatrix}. \tag{D7}
$$

The Ekman angle standard deviation is then taken as follows:

$$
\sigma_{\theta_E} = \min\left(\sigma_\theta, \pi\right). \tag{D8}
$$

*Author contributions.* SF performed the major part of the research and the writing of the manuscript. LU and AW contributed to the research and the writing.

*Competing interests.* The contact author has declared that none of the authors has any competing interests.

*Acknowledgements.* The forecasts, analyses, and related services are based on data and products of the Regional Center for Environmental
Modelling (CRMA), which is a sector of the Friuli Venezia Giulia Environmental Agency (ARPA FVG) ITALY (https://www.arpa.fvg.it, last access: 29 October 2024). The Piran HFR station has been operated by the Slovenian Environment Agency since July 2022. Between 2015 and July 2022 it was operated by the National Institute of Biology. ChatGPT was used for text refinement of Sect. 5.



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
