# Peer review of "Comparing an idealized deterministic-stochastic model (SUP model, version 1) of the tide-and-wind driven sea surface currents in the Gulf of Trieste to HF Radar observations"

_EGUsphere, 2024_

## Author Comment (AC1)

Dear Editor,

We are grateful to both reviewers for their corrections and comments as they have increased the quality of the paper.

Please find our detailed answers (written in blue) and corrections to both reviewers comments (reproduced in black) below. The corrections performed to the manuscript are given in red and an updated version of the manuscript with the corrections highlighted in red is provided.

Sincerely,

Sofia Flora, Laura Ursella and Achim Wirth

Reviewer 1

The paper introduces a hierarchy of idealized models to simulate surface current observations from an HFR site in northern Adriatic. They show that stochastic part of the signal (ie. residual current after tide and Ekman removal) obeys fat-tailed statistics with extremes occuring more often than expected from a Gaussian distribution. They employ deterministic, Gaussian and superstatistical stochastic models under several forcing protocols and observe the reproduction of statistical moments of each combination. The paper is interesting and might be of interest after some clarifications.

Perhaps i am missing something but in my opinion it would be interesting to take an existing hourly predictions from the CMEMS MFS model (or another full physics numerical model at your disposal, perhaps one with higher (r,t) resolution) as a baseline to compare your models to. They are hourly resolution but still it would be interesting to see what kind of statistics remains in the stochastic component from that full physics model.

Our work is about comparing an idealised model to observations to further our understanding of the processes involved and to evaluate their importance. Comparing observations to a more involved numerical model is very interesting but not the subject of the present work. It would involve (besides other points) a detailed discussion of the bulk formulas representing air-sea fluxes and to evaluate their impact. Such research must also include a strategy of how to improve the bulk formulas based on the idealized model and the observations. Such paper is very different from the research we performed and the paper we wrote.

We added in Sect. 5:

"Additionally, evaluating the air-sea interaction bulk formulas in CMEMS MFS model (or another full physics numerical model adopted for the Gulf of Trieste) in light of our observations, and therefore of the SUP model findings, would be extremely interesting.".

Specific remarks:

L156: you mention $\gamma_u$ is "adjusted to obtain the observed decay function" - how is it adjusted, through what method?

The $\gamma_u$ coefficient rules the temporal decay of the stochastic sea surface velocity $u_S$ ($v_S$) and it then contributes to the temporal decay of the modelled total sea surface velocity $u_o$ ($v_o$) correlations. The $\gamma_u$'s numerical value is then fixed in order to obtain, for the SUP $u_o$ auto-correlation function (Fig. 7, continuous red line), the same initial observed decay (Fig. 7, black asterisks). Since an analytical solution of $u_o$ is not available, it is not possible to apply a least square methodology. We then tested different values of $\gamma_u$, finding the most appropriate one.

We modified in Sect. 3.1:

"$\gamma_u$ is a coefficient which is a posteriori adjusted to obtain the observed HFR initial decay of the autocorrelation function of the SUP model sea surface currents (Fig. 7)".

Eq 4: $\nu$ should be defined already here, not later in the text

We modified in Sect. 3.1:

"In the SUP model, the $Q$ terms originate themselves from $2\nu$ stochastic processes $\alpha_i$: $Q = \sum_{i=0}^{2\nu} \alpha_i^2$. The variable $\nu$ gives the degrees of freedom (dof), according to the interpretation of Flora et al. (2023): the system has one dof if its positive variable

characterizing the variability of the system maximazes the entropy, i.e. it is exponentially distributed. This corresponds to half the value of Beck and Cohen (2003)'s interpretation. Each stochastic process $\alpha_i$ is governed by..."

and

"In the SUP model the variables $Q = \sum_{i=0}^{2\nu}\alpha_i^2$ depend on the variables $\nu$ and $\alpha_i$ , as shown in Table 1. ".

L 168: please introduce Langevin equation and Ornstein-Uhlenbeck process in broad terms or refer readers to the appendix. You describe Ekman and Coriolis effects in more detail while perhaps it should be the other way around.
We modified in Sect. 3.1:
"The variables $\alpha_i$ are solutions of the Langevin equation: they are Ornstein-Uhlenbeck processes (we refer the reader to Kloeden and Platen (1999) for a pedagogical discussion of stochastic processes), characterized by a Gaussian statistics".

L176: what would be a physical interpretation of $\nu = 1/2$?
The physical interpretation of $\nu$, the stochastic degrees of freedom, resides in the variability quantification of unresolved processes. Additionally, we adopted the following definition, according to Flora et al. (2023): the system has one dof if its positive variable characterizing the variability of the system maximazes the entropy, i.e. it is exponentially distributed. This corresponds to half the value of Beck and Cohen (2003)'s interpretation. So, $\nu = 1/2$ can be interpreted as a variability reduction (non-maximal entropy) in the unresolved processes, represented by the stochastic variables.

We added in Sect. 3.1:
"A decrease of the stochastic degrees of freedom $\nu$ leads to a reduction of the variability (non-maximal entropy) in the unresolved processes, represented by the stochastic variables.".

L190: unknow = typo
Corrected, thank you.

L191: how is $\beta$ increased empirically? What does this mean?
Similarly to the case of $\gamma_u$, we adjusted the numerical value of $\beta$ through some tests, in order to fit the observed variance in the total velocity increment PDF.

We modified in Sect. 3.1:
"For this reason in this case the $\beta$ coefficient is increased empirically , in order to fit the observed variance in the total velocity increment PDF".

L219: why is M=1311?
M=1311 because it is the result of the integer division between the observation time (from 01/01/2021 to 18/10/2022) and the perturbation time $\Delta t = 12$ h (larger than the correlation time of 5 h). It is then the maximum number of times it is possible to apply the pertrubation and the FRR diagonal mean response addend calculation.

We modified in Sect. 3.2:
" The perturbation time is fixed to $\Delta t = 12$ h, allowing us to repeat the procedure $M = 1311 \gg 1$ times, and $\Delta u_S = \Delta v_S = 8$ cm/s".

L238-9: the way they are defined, are $\xi$ and $\epsilon$ correlated or not?
They have a common term and are both measuring the distance of the perturbed system from "something else" (i.e. the unperturbed system for $\xi$ and the observations for $\epsilon$). We could not find any mathematical connection between them.

L243: i am having trouble understanding exactly how this is done in the model? What do the forcing files look like for

example? HOW do you impose this forcing? And furthermore, how exactly is FP1 forcing enforced?

The models require the tidal and wind forcing inputs, that are deterministic, and so they are time series, i.e. sequences of data numerical values equally-spaced in time. The tidal input is obtained through a linear combination of the S1, M2 and S2 tidal components, while the wind time series is obtained from the WRF model forcastings (see Sect. 2), as described few lines after. These time series are then considered in the model time integration process when called from the equations of the models. In FP1 the forcing inputs are time series of constant zeros.

We modified in Sect. 4:

"We test the different models imposing different Forcing Protocols (FPs) in the tidal and wind  time series (Table 2). In the most complete FP the complete tidal and wind forcing is included (FP9), in the other FPs the forcing signal is either averaged by a moving average over 12 h or completely  set to zero".

L269; when you write the DET model is "trivial", do you mean zero?

Yes, exactly.

We modified in Sect. 4.1:

"The DET model is therefore  null, while the GAU and SUP models have a  non-zero stochatic part".

I think the entire paper would benefit from considering CMEMS MFS, for example showing it on Figure 3.

Evaluating the air-sea interaction bulk formulas in CMEMS MFS model (or another full physics numerical model adopted for the Gulf of Trieste) in light of our observations (and SUP model) would be extremely interesting. But we think it is not the aim of the present work, since here we have obtained an idealized model reproducing the statistics of the observations, directly. Comparing the statistics of the CMEMS MFS model (or another full physics numerical model) with the observed one can be the object of another interesting paper.

As for Figure 3: you mention that DET simulation is missing the negative peak on April 3rd, which is clearly visible in HFR obs. But i don't see the negative peak in GAU or SUP simulations either. Am I missing something? What should I be paying attention to? Please explain this to the readers.

You are right, none of the displayed modelled realizations of the sea surface currents time series mimic the observed peak, but it is normal. If our model were perfect, the observations would be one of the walkers and therefore would at times be outside the stochastic one-standard-deviation band, as are our two walkers plotted in the chart. This is explained in detail in Eq. (12)-(13), where numbers show very good agreement of the time spent inside one-standard-deviation band between observations and the statistics of a typical walker, and corresponding text.

We added in Sect. 4.1:

"We emphasise that if our model were perfect, the observations would be one of the walkers and therefore would at times be outside the SUP total velocity ensemble standard deviation $\sigma_{u_o}$ (red) band, as is the case for the two random walkers (green lines) reported in Fig. 3.".

Figure 5: again, is it possible to add a PDF from CMEMS MFS stochastic residual current to this otherwise interesting plot?

It would be extremely interesting, but not the aim of the present work, as explained above.

P32: typo in the first line

Corrected, thank you.

Figure A1 is unreadable...

The legend has now proper labels, thank you.

145 Reviewer 2

In this manuscript, the authors develop deterministic and stochastic parameterizations to represent the tide- and wind-driven sea surface currents in the Gulf of Trieste. A hierarchy of idealized models is introduced, where deterministic components account for tidal and Ekman forcing, while stochastic components, based on Langevin and superstatistical formulations, simulate unre-
150 solved small-scale variability. The model coefficients are calibrated to reproduce the Probability Density Functions (PDFs) of observed High Frequency Radar (HFR) velocity increments. The proposed models are evaluated under various wind and tidal forcing protocols and validated against observed statistics. The authors show that the full stochastic model (SUP) accurately reproduces the observed fat-tailed PDFs, captures extreme events, and improves the temporal autocorrelations. Furthermore, they demonstrate that when the complete tidal and wind forcing is used, the stochastic component requires fewer degrees of
155 freedom, as more of the dynamics are captured deterministically.

This paper is well written, clearly structured, and presents compelling results. Overall, I recommend it for publication after minor revisions. Below I provide several comments aimed at further improving the manuscript.

160 The conclusion section would benefit from being more concise. In its current form, it is overly long and technical, which may obscure the key takeaways of the study. The authors are encouraged to focus on the most important messages that readers should retain from the results presented. Detailed discussions and technical aspects would be more appropriately placed in Section 4, if not already covered there. Alternatively, the authors could consider summarizing the main findings at the end of each subsection in Section 4 to enhance clarity.
165 We have removed the most technical aspects.

We modified in Sect. 5:
"
170 "

and

"In our case the variance is obtained through the sum of $2\nu$ squared (Gaussian) Ornstein-Uhlenbeck processes and results
175 to be Gamma distributed ."

and

"The velocities temporal auto-correlation functions under the FP9 (Fig. 7)
180   reveals that the SUP model, after a realistic time decay of around 5 h (driven by the deterministic signal), allows for a better representation  with respect to the DET model, of the temporal modulations.  The Ekman dynamics analysis (Fig. 9),
185   indicates that strong wind increases the likelihood of the SUP model generating trajectories that match the observed average Ekman state."

and
190

" The FRR holds when the stochastic signal is perturbed, showing that the response to an external perturbation can be obtained by considering the fluctuations of the unperturbed system. In particular, the SUP model response function decays with the stochastic velocity drag time scale ($1/\gamma_u$, see Fig. 11 and
195 Eq. (16)), indicating a clear memory time scale for external stochastic perturbations."

In the context of the stochastic models GAU and SUP, is there a specific justification for assuming that the components x and y are uncorrelated, as indicated by the diagonal covariance matrix in equation (2)? Could this assumption be relaxed to allow for a correlation between the two components of the stochastic velocity in (4)? A more thorough clarification of this
200 choice and its potential implications for the model would be appreciated.
The justification for assuming that the components x and y are uncorrelated is that we adopt the simplest possible approach reproducing the observed statistics. It is possible to consider correlated x and y (and so correlated $u_S$ and $v_S$), but not necessary.

We added in Sect. 3.1:
205 "The stochastic velocity, representing the fast unresolved turbulent dynamics and present in the GAU and SUP models, is defined by the following set of equations (for the purpose of the study, i.e. to reproduce the observed superstatistical statistics, it is sufficient to have uncorrelated stochastic velocity components): [...]".

Line 298: The authors mention that data assimilation methods are adopted, but it is unclear which specific techniques are
210 employed. Are they using 4DVar, EnKF, nudging, or another method? If this has already been addressed and I have overlooked it, I apologize. Otherwise, I would appreciate a clarification.
We apologize for the incorrect terminology, we used a perturbation method where, every time window $\Delta t$ we have perturbed the model system with the observations, which is supposed to be without error. Such experiment is important to estimate the predictability of the dynamics and therefore also the ability of data assimilation.

215

We modified in Sect. 3.3:
"In order to test predictability capabilities of the SUP model,  perturbation methods are adopted  assuming that the HFR observations are not affected by observational uncertainty and represent the reality, i.e. modelled data are replaced by observations. In the following this method is called "observation-based perturba-
220 tion". In detail, every $\Delta t$ time the perturbed system ($\boldsymbol{u_{o,P}}$) is updated to the observed HFR velocities (both the $u_o$ and $v_o$ components in the same simulation), in particular it is the stochastic signal to be perturbed."

and in Sect. 4.3:
"In Fig. 12 the  observation-based perturbation methodology on the time series, described in Sect. 3.3, and
225 the effects on the total sea surface velocities are visualized. [...] Both the total SUP $u_o$ and $v_o$ show the  observation-based perturbation subject to an evident decay in time."

and

230 "The external perturbation, due to the  observation-based methodology, decays with the stochastic drag coefficient when we perturbe the stochastic velocity, while a clear correspondance with the drag coefficient is not found when the Ekman signal is perturbed."

and in Sect. 5:
235 "The  predictability evaulation through the observation-based perturbation shows that the SUP model, given any external correction to the HFR values, reaches an equilibrium distance from the observations, converges to the unperturbed realization, after 5 hours (Fig. 13)."

and in Code and data availability:
240 "Additionally, `twb_FRRD_model.f90`, `twb_FRRS_model.f90`, `twb_datassD_model.f90` and `twb_datassS_model.f90` are the codes implementing the STO model with $\nu = 1/2$ in the FRR and  observation-based perturbation applications (applied to the deterministic or stochastic signals).".

The calculations presented in the appendices could be streamlined. In particular, several equations—such as (A4), (A13), (A16), (A19), (A20), (A22–A24), (A27), (A31), (B8), (C5–C7), (D4), and (D7)—include intermediate steps that could be omitted, especially where the variables involved have already been introduced earlier in the manuscript. To enhance clarity and conciseness, the authors are encouraged to reduce unnecessary multi-line derivations and express equations in a single line where feasible.

Regarding Appendix A:

Nowadays scientists are less trained to do analytical calculations. Maintaining the derivations helps to speed up the reading and contributes to the demystification of stochastistic calculus.

Regarding Appendix B, Appendix C and Appendix D:

Eq. (B5) and Eq. (B8) are now in one line;

Eq. (C6)

$$\begin{cases} \partial_t u_E = -\frac{\alpha}{h}|\boldsymbol{u_E}|u_E + \zeta|\boldsymbol{u_E}|u_E + f v_D + \frac{F_u}{h} \\ \partial_t v_E = -\frac{\alpha}{h}|\boldsymbol{u_E}|v_E + \zeta|\boldsymbol{u_E}|v_E - f u_E + \frac{F_v}{h} \end{cases}$$

is now removed;

Eq. (D4) and Eq. (D7) have less intermediate steps.

Typos:

Line 69: Lorentz models -> Lorenz system

Line 70: hierachy -> hierarchy

Line 118: paramerterized -> parameterized

Line 175: the bracket notation <Q> should be introduced here not later in Line 215

Line 288: currrents -> currents

Line 293: istantaneous -> instantaneous

Lines 318, 347: occurrance -> occurrence

Line 418: normailzed -> normalized

Line 424: correspondance -> correspondence

Line 646: respect -> with respect to, semplicity -> simplicity

Line 659: indipendent -> independent

Table 4 caption: (Flora et al., 2023) -> Flora et al. (2023)

Corrected, thank you. We replaced $\langle Q \rangle$ with $E[Q]$, since it indicates the first moment of the $Q$ variable in the SUP model framework.

Suggestions:

Line 74: Our modelization -> Our modelling approach

Line 78: wind-and-tide-driven circulation -> wind- and tide-driven circulation

Table 4: The authors may consider adding brief descriptions of the listed parameters to enhance clarity and understanding—for example, water density $\rho$. More importantly, it would be helpful for the model-specific parameters.

Line 325, 326: due probably to -> probably due to, deviate slightly -> slightly deviate

Corrected, thank you.

**References**

285   Beck, C. and Cohen, E. G.: Superstatistics, Physica A: Statistical mechanics and its applications, 322, 267–275, 2003.

Flora, S., Ursella, L., and Wirth, A.: Superstatistical analysis of sea surface currents in the Gulf of Trieste, measured by high-frequency radar, and its relation to wind regimes using the maximum-entropy principle, Nonlinear Processes in Geophysics, 30, 515–525, 2023.

Kloeden, P. E. and Platen, E.: Numerical Solution of Stochastic Differential Equations, Springer, 1999.